



# Modeling the response of soil moisture to climate variability in the Mediterranean region

Louise Mimeau[1,2], Yves Tramblay[1], Luca Brocca[3], Christian Massari[3], Stefania Camici[3], and Pascal Finaud-Guyot[1,4]

[1]HSM (Université de Montpellier, CNRS, IRD), Montpellier, France
[2]Departamento de Ingeniería Civil, Facultad de Ciencias Físicas y Matemáticas, Universidad de Chile, Santiago, Chile
[3]Research Institute for Geo-Hydrological Protection, National Research Council, Perugia, Italy
[4]INRIA Lemon

*Correspondence to:* Yves Tramblay (yves.tramblay@ird.fr)

**Abstract.** Future climate scenarios for the Mediterranean region indicate a possible decrease in annual precipitation associated with an intensification of extreme rainfall events in the coming years. A major challenge in this region is to evaluate the impacts of changing precipitation patterns on extreme hydrological events such as droughts and floods. For this, it is important to understand the impact of climate change on soil moisture since it is a proxy for agricultural droughts and the antecedent soil

moisture condition plays a key role on runoff generation. This study focuses on 10 sites, located in Southern France, with available soil moisture, temperature, and precipitation observations on a 10 year time period. Soil moisture is simulated at each site at the hourly time step using a model of soil water content. The sensitivity of the simulated soil moisture to different changes in precipitation and temperature is evaluated by simulating the soil moisture response to temperature and precipitation scenarios generated using a delta change method for temperature and a stochastic model (Neyman-Scott rectangular pulse model) for

precipitation. Results show that soil moisture is more impacted by changes in precipitation intermittence than precipitation intensity and temperature. Overall, increased temperature and precipitation intensity associated with more intermittent precipitation leads to decreased soil moisture and an increase in the annual number of days with dry soil moisture conditions. In particular, a temperature increase of +4 °C combined with a decrease of annual rainfall between 10 and 20 %, corresponding to the current available climate scenarios for the Mediterranean, lead to a lengthening of the drought period from June to October

with in average +22 days of soil moisture drought per year.

## 1 Introduction

The Mediterranean region is a transitional zone between dry and wet climates and in these semi-arid areas the direct evaporation from the soil plays an important role on the surface energy balance, with evapotranspiration strongly dependent on available soil moisture (Koster et al., 2004; Seneviratne et al., 2010; Taylor, 2015). Consequently, the Mediterranean has been identified

as a region with a strong coupling between the atmosphere and the land surface, with feedback effects of soil moisture on temperature and also precipitation (Seneviratne et al., 2010; Knist et al., 2017; Hertig et al., 2019). Indeed, soil moisture is a key variable in the hydrological cycle for the partitioning of rainfall into infiltration and runoff and also for the mass and



energy balance between land surface and the atmosphere (Seneviratne et al., 2010; Brocca et al., 2017). The water contained in the unsaturated, or vadose zone, is an important driver for floods with soils close to saturation having more probability to produce runoff when subjected to precipitation inputs (Zehe et al., 2005; Ivancic and Shaw, 2015; Woldemeskel and Sharma, 2016; Bennett et al., 2018). This is particularly true in the Mediterranean context where several studies have shown the strong

influence of soil moisture on flood generation processes (Brocca et al., 2008; Penna et al., 2011; Tramblay et al., 2010; Uber et al., 2018; Tramblay et al., 2019). Similarly, the soil moisture is an important parameter for drought analysis, since low soil moisture content is a good proxy for drought impacts on agriculture or wildfires occurrence (Vidal et al., 2010; Ruffault et al., 2013).

There is a climatic trend towards a drying of the Mediterranean region, both during the historical period but also in future

climate scenarios, showing a decrease in precipitation amounts and occurrence, associated with an increasing frequency of drought episodes (Hoerling et al., 2012; Polade et al., 2014, 2017; Hertig and Tramblay, 2017; Lionello and Scarascia, 2018). For a RCP8.5 emission scenario, Giannakopoulos et al. (2009) and Polade et al. (2014) both estimate a mean decrease up to -30 % of the annual precipitation in the Mediterranean region by the end of the century and increase of dry days ranging between +1 to +3 weeks per year. For soil moisture, the precipitation decrease associated with higher temperatures leading

to stronger evaporation rates is causing a decrease in soil moisture for large parts of the Mediterranean (Vidal et al., 2010; Vicente-Serrano et al., 2014; Hanel et al., 2018). Samaniego et al. (2018) and Grillakis (2019) provided future projections of soil moisture for Europe using different combinations of climate scenarios from General Circulation Models (GCM), Regional Climate Models (RCM), hydrological and land surface models, showing a clear climate signal towards a future decrease in soil moisture content and consequently increase in agricultural droughts for Mediterranean regions.

Only a few studies attempted to validate the soil moisture simulated by the GCM or RCM land surface schemes, probably due to the lack of sufficient networks with in situ soil moisture measurements, which show high spatial variability (Brocca et al., 2007; Crow et al., 2012). Yuan and Quiring (2017) validated ensemble of Coupled Model Intercomparison Project Phase 5 (CMIP5 GCMs) over North America with in situ and satellite soil moisture observations. Knist et al. (2017) evaluated the Coordinated Regional Climate Downscalling experiment (CORDEX RCM) over Europe using GLEAM (Global Land

Evaporation Amsterdam Model) and FLUXNET reference data. If the main patterns of seasonal soil moisture were found adequately represented from climate models in both studies, they also pointed out the large multi-model variability in particular in the transitional climate zones. Indeed, many studies reported a high model dependence of soil moisture simulations (Koster et al., 2009; Berg et al., 2017). This is particularly true for the Mediterranean regions, due to structural uncertainty, different process representation, soil depths and interactions with vegetation that are not currently adequately reproduced by land surface

models (Knist et al., 2017; Quintana-Seguí et al., 2019).

Beside the use of climate models, scenarios neutral approaches are increasingly employed to assess the vulnerability of water resources under different climate change scenarios (Prudhomme et al., 2010; Guo et al., 2017). The approach is similar to a sensitivity analysis aiming at quantifying the changes in a given hydrological variable for a plausible range of changes in hydrometerological conditions. Therefore, it can provide useful information to identify the hydro-meteorological parameters

that have the greatest impact on a given response variable. Guo et al. (2018) provided an example of such a scenario-neutral





approach based on a stochastic weather generator to explore possible rates of changes in rainfall intermittency and extremes in Southern Australia. The only study that applied this method to soil moisture, to the knowledge of the authors, is by Yoo et al. (2005) who coupled a stochastic generator of rainfall to a soil moisture model in the Walnut Gulch experimental watershed in southeastern Arizona to estimate soil moisture changes due to rainfall variability. They found that rainfall arrival rates was

the most sensitive parameter, with decreasing soil moisture content with increasing rain intermittence, even without a decrease of the total volume of rainfall. Yet, this type of approach needs to be applied to other land regions and different sites in order to evaluate the possible spatial variability in addition to the temporal variability of rainfall. These bottom-up approaches are complementary to the modelling chains linking climate and land surface models, and document the most relevant process leading to soil moisture changes than in turn can be used to improve the land surface schemes.

The objective of this study is to analyze the variability of soil moisture for a set of Mediterranean sites according to changes in precipitation and temperature. The method relies on the use of a stochastic precipitation generator coupled with the soil moisture model proposed by Brocca et al. (2008). The scientific questions addressed in the present work are: which precipitation characteristics (intermittency, intensity) do influence soil moisture changes, in conjunction with changes in temperature as a proxy for evapotranspiration changes? And how this response of soil moisture to changes in climate drivers varies in space for

a range of different locations with different topographical and soil properties?

The paper is structured as follows: Sect. 2 describes the study area and collected datasets; Sect. 3 provides a description of the soil moisture and stochastic rainfall models (Sect. 3.1 to 3.3), and of the experimental design for the simulation of the soil moisture scenarios (Sect. 3.4); Sect. 4 presents the validation of stochastic rainfall model (Sect. 4.1) and soil moisture model (Sect. 4.2) after calibration and the sensitivity analysis of the median (Sect. 4.3) and extreme soil moisture (Sect. 4.4) to

precipitation and temperature variations; Sect. 5 discusses the results and summarises the main conclusions of the paper.

## 2  Data

This study uses soil moisture, precipitation and temperature in situ data from 10 stations of the SMOSMANIA network (Calvet et al., 2007; Albergel et al., 2008) located in the French Mediterranean region (Fig 1).

Stations all present a characteristic Mediterranean precipitation seasonal cycle with a hot and dry summer followed by heavy

precipitation between September and November (Fig 2). This precipitation cycle directly impacts soil moisture with lower soil moisture values during summer and higher values during winter. Although all stations are located in the French Mediterranean region, they differ in altitudes, ranging from 30 (Pezenas) to 538 m.a.s.l. (Mouthoumet), in mean annual precipitation, ranging from 500 mm (Lézignan Corbières, Pézenas) to 1734 mm (Barnas), and in soil characteristics (Table 1). The station altitude is correlated to mean annual precipitation (r=0.7), except for the station Mouthoumet with lower annual precipitation than stations with comparable altitude (if this station is removed, r=0.92 between altitude and mean annual precipitation).

In-situ data is collected at hourly time step and covers a period starting in 01/01/2007 to 31/12/2016. Soil moisture data series used in this study are computed from measurements at four different depths (5, 10, 20, and 30 cm) as the weighted average as



a function of soil layer depth. The integration of the measurements at various depths enables to have a representation of the average soil moisture in the root zone layer.

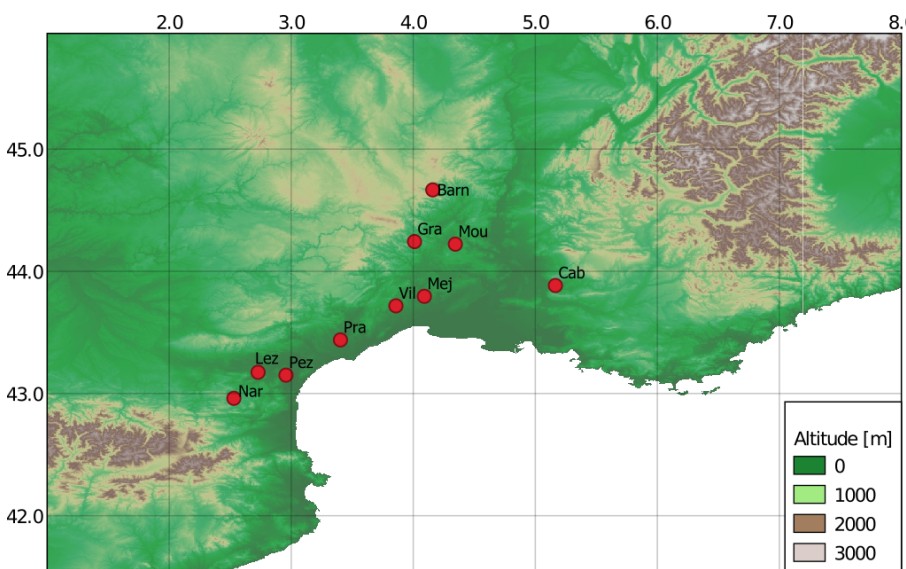

**Figure 1.** Localisation of the study sites in southern France.





| Code | Name | Lat [°] | Lon [°] | Altitude [m] | Precipitation [mm/yr] | Soil properties | | | | Landcover |
|------|------|---------|---------|--------------|-----------------------|------|------|------|----------------|-----------|
| | | | | | | Clay [%] | Sand [%] | Silt [%] | Soil class (ISSS) | |
| Barn | Barnas | 44.666 | 4.16 | 480 | 1734 | 9.5 | 77.3 | 13.2 | Sandy loam | Tree cover |
| Cab | Cabrieres d'Avignon | 43.884 | 5.165 | 142 | 697 | 24.2 | 47.6 | 28.2 | Clay loam | Cropland |
| Gra | La Grand Combe | 44.243 | 4.01 | 499 | 1412 | 12.9 | 73.2 | 13.9 | Sandy loam | Urban areas |
| Lez | Lezignan Corbieres | 43.173 | 2.728 | 60 | 502 | 27.3 | 44 | 28.7 | Light clay | Urban areas |
| Mej | Mejannes-le-Clap | 44.222 | 4.345 | 318 | 992 | 16.2 | 45.5 | 38.3 | Clay loam | Grassland |
| Mou | Mouthoumet | 42.96 | 2.53 | 538 | 689 | 29.4 | 42 | 28.6 | Light clay | Grassland |
| Nar | Narbonne | 43.15 | 2.957 | 112 | 530 | 46.4 | 26.2 | 27.4 | Heavy clay | Cropland |
| Pez | Pezenas | 43.438 | 3.403 | 30 | 508 | 17.5 | 50.6 | 31.9 | Clay loam | Cropland |
| Pra | Prades-le-Lez | 43.717 | 3.858 | 85 | 816 | 31.1 | 27 | 41.9 | Light clay | Cropland |
| Vil | Villevielle | 43.795 | 4.091 | 41 | 756 | 13.6 | 65.7 | 20.7 | Sandy loam | Cropland |

**Table 1.** Stations characteristics

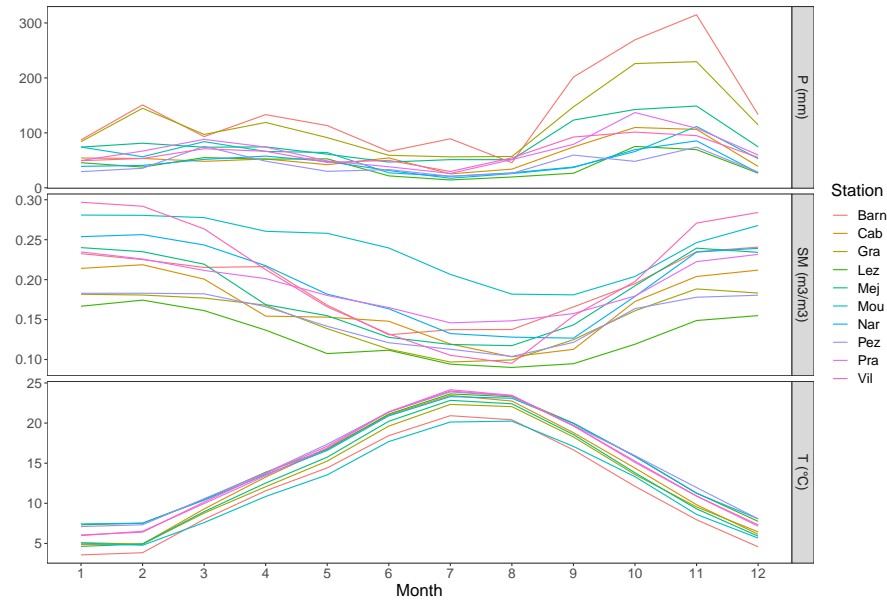

**Figure 2.** Observed seasonal cycle of precipitation, soil moisture, and air temperature at stations.





## 3  Method

### 3.1  Soil moisture model

The soil moisture model developed by Brocca et al. (2008) is used to simulate present soil moisture and soil moisture response under different climate scenarios. The soil moisture model (SMmodel) incorporates a Green-Ampt approach for infiltration, a gravity-driven approximation for drainage, and a linear relationship between actual and potential evapotranspiration as a function of soil moisture. The SMmodel simulates the hourly temporal evolution of soil moisture and actual evapotranspiration. Hourly precipitation and air temperature are used as input into the SMmodel, potential evapotranspiration is computed from air temperature through the Blaney and Criddle approach. Details on the model equations can be found in Brocca et al. (2008) and Brocca et al. (2014). The model has been applied at multiple sites in Italy and Europe (e.g., Brocca et al. (2014)) with satisfactory results.

### 3.2  Soil moisture model calibration

The SMmodel uses fixed and calibrated parameters. The fixed parameters values (Table 2) were estimated based on the observed soil moisture and geographic location of the stations. Three parameters were calibrated: hydraulic conductivity $K_s$, root zone depth $Z$, exponent of drainage $m$, and coefficient for evapotranspiration $K_c$ (calibration ranges in Table 2). These parameters are calibrated for each station using the total period of observed data, but two additional calibrations were performed on sub-periods (first half and second half of the total period) in order to analyze the stability of the calibration. For the calibration process, missing precipitation and temperature data are reconstructed by replacing missing precipitation with an intensity of 0 mm/hr, and by linearly interpolating temperature data for gaps of less than 3 hours or using the climate mean otherwise. Time steps with reconstructed precipitation and temperature are not taken into account in the calculation of the NSE coefficient used as optimization criterion for the calibration (Nash and Sutcliffe, 1970). Details on missing data at each stations are presented in Annexe 4.





| Fixed parameter | Value |
| --- | --- |
| Wetting front soil suction head $\psi$ | -155.0 mm |
| Initial condition $\theta_0$ | 0.2 m$^3$/m$^3$ |
| Saturated soil moisture $\theta_{sat}$ | max of observed soil moisture |
| Residual soil moisture $\theta_{res}$ | min of observed soil moisture |
| Monthly potential evaporation coefficient $L$ | 0.208 0.234 0.266 0.300 0.329 0.345 0.339 0.314 0.282 0.248 0.218 0.201 |
| Soil layer depth $Z$ | 300 mm |
| Calibrated parameter | Range |
| Hydraulic conductivity $K_s$ | 0.01 $<K_s<$100 mm/h |
| Exponent of drainage $m$ | 1 $<m<$45 |
| Evaporation coefficient $K_c$ | 0.5 $<K_c<$2 |

**Table 2.** Fixed parameters values and ranges of calibrated parameters of the soil moisture model. Fixed parameter $L$ is calculated as the monthly percentage of total daytime hours out of total daytime hours of the year.

### 3.3 Generation of temperature and rainfall scenarios

For each station, a 20 years temperature data serie is generated by repeating the hourly climatic mean. Temperature scenarios are generated by applying a delta of ranging between +0°C and +4°C.

The stochastic weather generator, Neyman-Scott rectangular pulse model, NSRP (Cowpertwait et al., 1996) is used to gen-
5 erate 20 series of hourly rainfall data time series. The peculiarity of the model lies in its capability to preserve the statistical properties of benchmark rainfall time series over a range of time scales. As the model has been extensively described in previous papers (e.g. Cowpertwait et al., 1996; Camici et al., 2011) here only a brief discussion is given.

The NSRP model has 5 parameters:

– $\lambda$: mean waiting time between adjacent storm origins [hr].

– $\beta$: mean waiting time between raincell origins after storm origins [hr],

– $n$: mean number of raincell per storm,

– $\eta$: mean duration of raincell [hr],

– $\xi$: mean intensity of raincell [mm/hr]

A Poissonian process with parameter $\lambda$ controls the generation storm origins. For each storm origin, a random $n$ number
of raincell origins are generated displaced from the storm origin according to a $\beta$ parameter exponentially distributed process. Duration and intensity of each raincell are expressed by two other independent random variables assumed exponentially dis-
tributed with parameter $\eta$ and $\xi$ respectively. These parameters are estimated, for each month of the year, by minimizing an





objective function evaluated as the weighted sum of the normalized residuals between the statistical properties of the observed time series and their theoretical expression derived from the model.

As studies on future precipitation patterns in the Mediterranean region predict an increase of dry days frequency associated with an intensification of extreme precipitation events (Paxian et al., 2015; Polade et al., 2017; Tramblay and Somot, 2018), we
generate precipitation scenarios with increasing precipitation intermittence and increasing mean intensity by applying deltas from +0 to +50 % on $\lambda$ and $\xi$ parameters (see details in Sect.3.4). For each precipitation scenarios, 20 precipitation data series are generated with the NSRP model over a 20 years period and used as input of the soil moisture model.

## 3.4 Sensitivity analysis of the simulated soil moisture to precipitation and temperature changes

### 3.4.1 Direct analysis

We first analyze the sensitivity of the simulated soil moisture for specific changes in temperature and precipitation. We consider three temperature scenarios with $\Delta T = +0, +2,$ and $+4C$, and 121 precipitation scenarios with $\Delta \xi$ and $\Delta \lambda$ regularly spaced between +0 and +50 % with a 5 % step. The soil moisture model is then run for each precipitation and temperature scenarios (i.e. 363 scenarios per stations) to analyze the sensitivity of the simulated soil moisture to temperature and precipitation changes. Figure 3 resumes the process for the simulation of the soil moisture scenarios. The simulation with no change in temperature
and precipitation intensity and intermittence is called the reference scenario and is used to represent soil moisture conditions under present climate. The evolution of extreme soil moisture events is evaluated by estimating the mean number of days per year under soil water excess, and drought. We consider episodes of soil water excess as consecutive days with a daily soil moisture above the reference scenario 95[th] percentile, and drought episodes as days with soil moisture below the 5[th] percentile. Considering the modeling chain as *(i)* the NSRP model (depending on the calibrated values of $\beta, \nu, \eta, \lambda$ and $\xi$ and the applied
perturbations $\Delta \lambda$ and $\Delta \xi$); *(ii)* the temperature scenario generation perturbed of $\Delta T$ and *(iii)* the SM model, for a given set of parameters, the modeling chain is processed 20 times. Quantiles and annual numbers of days under drought or soil water excess are computed for each of the 20 corresponding soil moisture results and then averaged to produce a unique scenario.



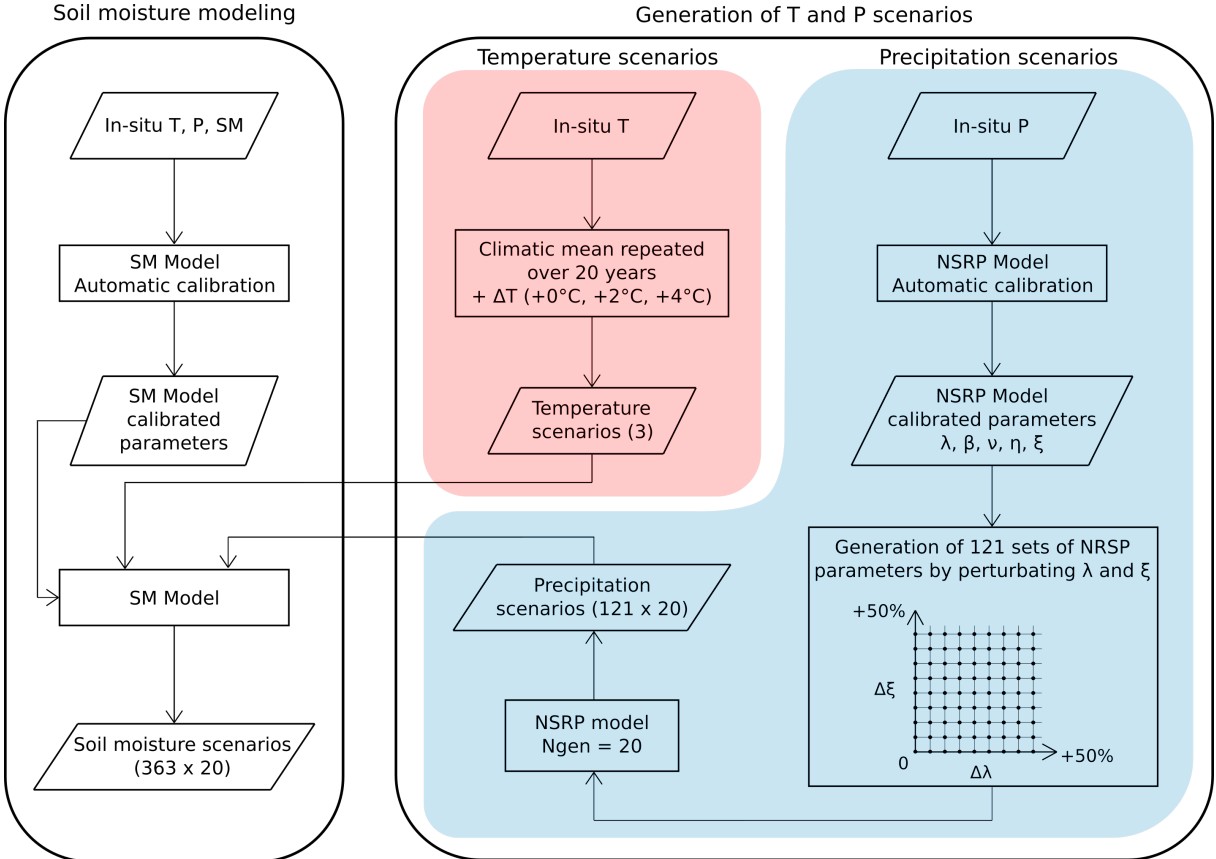

**Figure 3.** Flowchart of the experimental design for the simulation of the soil moisture scenarios

### 3.4.2 Global Sensitivity Analysis

A Global Sensitivity Analysis (GSA) (Saltelli et al., 2008; Pianosi et al., 2016) assess the model behavior (model output sensitivity to the input parameters) in the whole parameter space using a variance decomposition method. Considering $Y = f(\underline{X})$ with $Y$ the output of the model $f$ to a set of parameters $\underline{X} = (X_1, X_2, ..., X_N)$. A functional ANOVA decomposition is

5   applied to $Y$ (e.g. (Sobol, 1993; Saltelli et al., 2010)) :

$$V(Y) = \sum_{i=1}^{N} V_i + \sum_{i=1}^{N} \sum_{j>i}^{N} V_{i,j} + ... + V_{1,2,...,N}$$

where $N$ represents the number of sampled parameters. $V(Y)$ is the total variance of the model output, $V_i$ the first order variance of $Y$ due to parameter $X_i$, $V_{ij}$ the second order variance (covariance) of $Y$ due to $X_i$ and $X_j$ and the ) and higher order variance due to more than 2 parameters. A first-order Sobol index $S_i$ corresponds to the ratio of the corresponding

10   variance $V_i$ to the total variance $V(Y) : S_i = V_i/V(Y)$ and is thus always between 0 and 1. The sum of all the (first and higher order) Sobol indices is equal to unity.





Assuming that the changes in temperature and precipitation are stochastic variables, the first-order Sobol indices are computed using the state dependent parameter modelling proposed by (Ratto et al., 2007). For the Global Sensitivity Analysis, a different set of 1000 sets of temperature and precipitation changes, generated randomly in the range of values presented in section 3.3, is used in order to cast continuously the range of values ($\Delta T = [+0; +4\ ^{\circ}C]$, $\Delta\lambda = [0; 50\%]\,\lambda$ and $\Delta\xi = [0; 50\%]\,\xi$).

The objective of this sensitivity analysis is to estimate the relative influences of changes in temperature and precipitation characteristics on soil moisture.

## 4   Results

### 4.1   NSRP model calibration and generated rainfall scenarios

Rainfall series generated with NSRP model for the reference scenario show good agreement with the observed rainfall charac-

teristics. Figure 4a shows that the mean annual amount of rainfall is well reproduced by the model (r2=0.99) and that the range of values of annual amount of rainfall is also comparable to the range of observed values. The mean annual number of dry days (i.e. days with precipitation below 1 mm) is similar to observations reproduced (r2=0.63) (Fig. 4b). NSRP model tends to slightly overestimate lower values of the daily intensities distribution (Fig. 4c), but overall, the simulated distributions are in good agreement with observed distributions.

The perturbation of the NSRP parameters for the mean intensity $\xi$, and for the rainfall intermittence $\lambda$, from +0 to +50 %, enables to produce rainfall scenarios with different patterns. An increase of +50 % of the rainfall mean intensity with an unchanged intermittence leads to a mean increase of the annual rainfall of 432 mm associated with an increase of mean rainfall intensity of wet days of +4.8 mm/day. On the opposite, an increase of +50 % of the rainfall intermittence with an unchanged mean intensity leads to a mean decrease of of the annual rainfall of -350 mm (35 % of original annual rainfall) and an increase

of +23 days/yr of dry days. An increase of +50 % of both parameters leads to an unchanged mean annual rainfall but with an increase of mean rainfall intensity of +4.0 mm/day and an increase of +20 days/yr of dry days.





**Figure 4.** Characteristics of simulated rainfall with NSRP model for the reference scenario and observed rainfall. Comparison of simulated and observed a) annual rainfall b) annual number of dry days (dots represent mean values and bars the range from minimal to maximal simulated or observed values). c) Q-Q plot of daily rainfall intensities (dots represent deciles values).



## 4.2 SM model calibration

Table 3 presents the calibrated parameters of the SMmodel and NSE values after calibration. Calibrated values of $K_s$ are consistent with the range of hydraulic conductivities for natural soils (Angerer et al., 2014) (Table 5). The NSE values for the calibration on the total period are all above 0.6, and 9 stations out of 11 have a NSE value above 0.75. RMSE values range from 0.015 to 0.032 $m^3.m^{-3}$. These results show that the SMmodel is able to simulate soil moisture accurately in the present period 2007-2016. Calibrations on the sub-periods (first and second halves of each station time series) lead to similar parameters (not shown here) and NSE values on both sub-periods, showing that the calibration is stable for the selected period. Lower NSE values for the calibration on sub-periods are due to missing observed data unevenly distributed over the total period.

|  | Barn | Cab | Gra | Lez | Mej | Mou | Nar | Pez | Pra | Vil |
|---|---|---|---|---|---|---|---|---|---|---|
| $K_s$ (mm.hr$^{-1}$) | 38.1 | 34.3 | 35.9 | 23.1 | 28.8 | 36.2 | 51.1 | 14.6 | 59.6 | 6.9 |
| $m$ | 17.6 | 15.6 | 10.9 | 14.1 | 16.4 | 23.0 | 15.9 | 12.8 | 11.89 | 38.2 |
| $K_c$ | 1.17 | 1.43 | 1.74 | 1.22 | 1.81 | 0.94 | 1.26 | 1.99 | 1.32 | 1.63 |
| NSE | 0.76 | 0.77 | 0.93 | 0.85 | 0.9 | 0.63 | 0.91 | 0.789 | 0.65 | 0.9 |
| NSE 1 | 0.6 | 0.72 | 0.86 | 0.87 | 0.80 | 0.69 | 0.87 | 0.31 | 0.64 | 0.87 |
| NSE 2 | 0.71 | 0.75 | 0.87 | 0.78 | 0.91 | 0.04 | 0.912 | 0.60 | 0.57 | 0.86 |

**Table 3.** Calibrated parameters of the SM model and NSE validation values. NSE 1 and NSE 2 are the validation coefficients while calibrating respectively on the first and second sub-periods of the in situ data series.

The calibrated parameters are then used to simulate soil moisture under different scenarios of temperature and precipitation. Figure 5 compares the distributions of observed daily soil moisture with simulated daily soil moisture forced with the reference scenario. The results show that in the reference scenario soil moisture distribution is in very good agreement with the distribution of observed soil moisture (except for Mazan-Abbaye station which seems to overestimate low values of the distribution). The bias between the mean soil moisture from the reference scenario and the mean observed soil moisture is low and ranging from -0.003 to 0.01 $m^3.m^{-3}$ for the different stations.





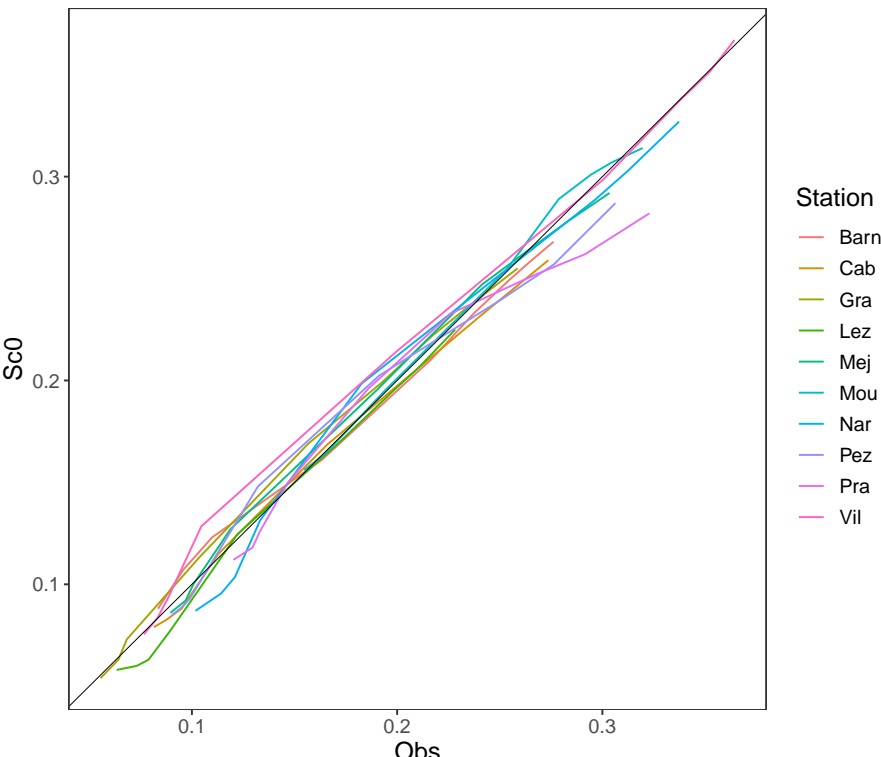

**Figure 5.** Q-Q plot of simulated (reference scenario) and observed daily soil moisture.

## 4.3 Sensitivity of soil moisture to precipitation and temperature changes

Figure 6 shows the sensitivity of the median of simulated soil moisture to changes in precipitation patterns. Results show that the median soil moisture is more sensitive to changes in precipitation intermittence ($\Delta\lambda$) than to changes in precipitation mean intensity ($\Delta\xi$). For the +0°C scenario, an increase of the precipitation intermittence of +50 % leads to a decrease between

5  -8 and -21 % on the median soil moisture, whereas an increase of 50 % in the precipitation mean intensity only leads to an increase of the median soil moisture ranging between +2 and +17 %. Results also show that stations have different sensitivity to precipitation and temperature changes. Stations such as Villevielle, Narbonne seems to be more sensitive to climate variability, whereas Barnas, La Grand-Combe, Mouthoumet and Prades-le-Lez stations show a lower impact of changing precipitation patterns and temperature on the median soil moisture. Figure 7 shows the correlation between the median soil moisture change to different scenarios with the stations climatic characteristics (observed annual mean temperature and precipitation). Results

10  show that the soil moisture response is correlated to the station local temperature and also to local precipitation to a lesser extent (Fig. 7). Southern stations presenting a warmer and dryer climate seem to be more impacted by changes in precipitation and temperature than northern stations located in the Cevennes mountain range with a colder and wetter climate. No correlation was





found between the soil moisture response and the NSRP model and SM model parameters values, meaning that the observed variability between station is independant from the models calibrations.

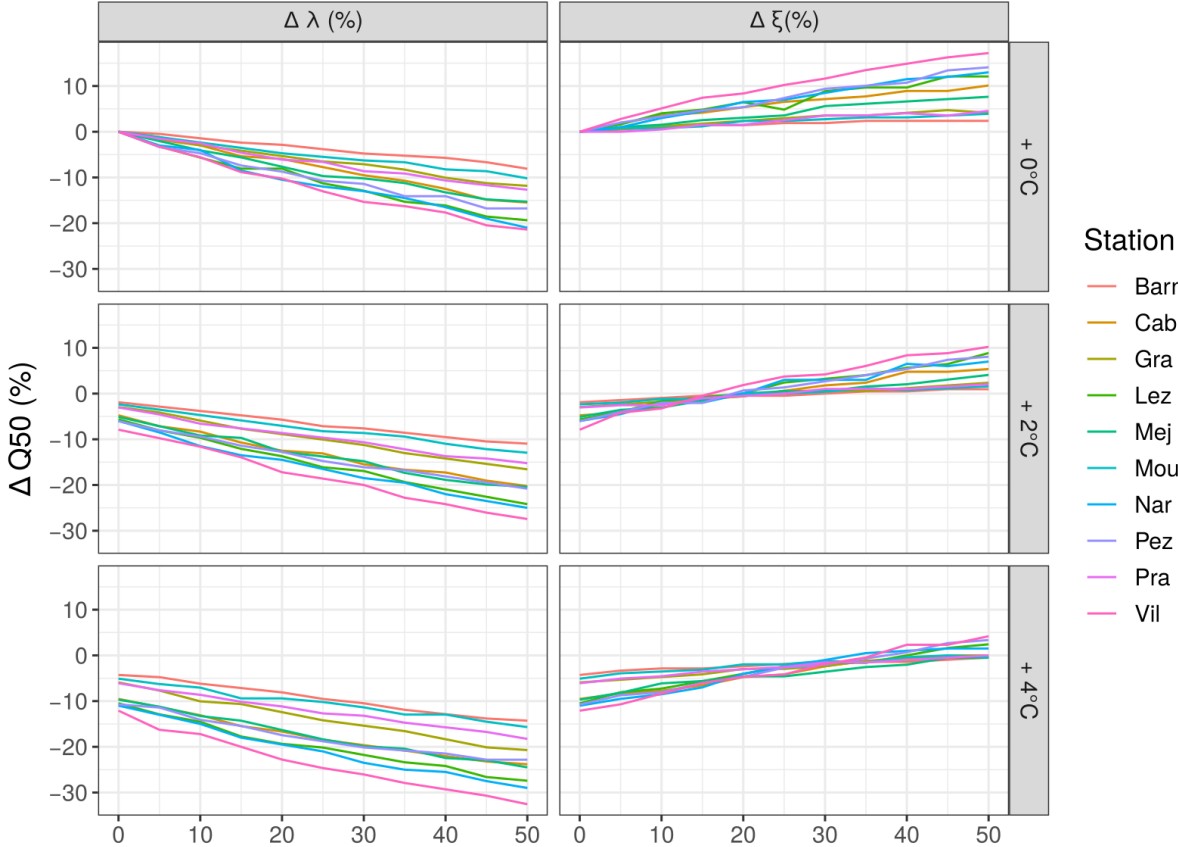

**Figure 6.** Sensitivity of the median of the simulated soil moisture to an increase of the precipitation intermittence (left panel), and to an increase of mean precipitation intensity (right panel) under different temperature scenarios (+0°C, +2°C, +4°C)





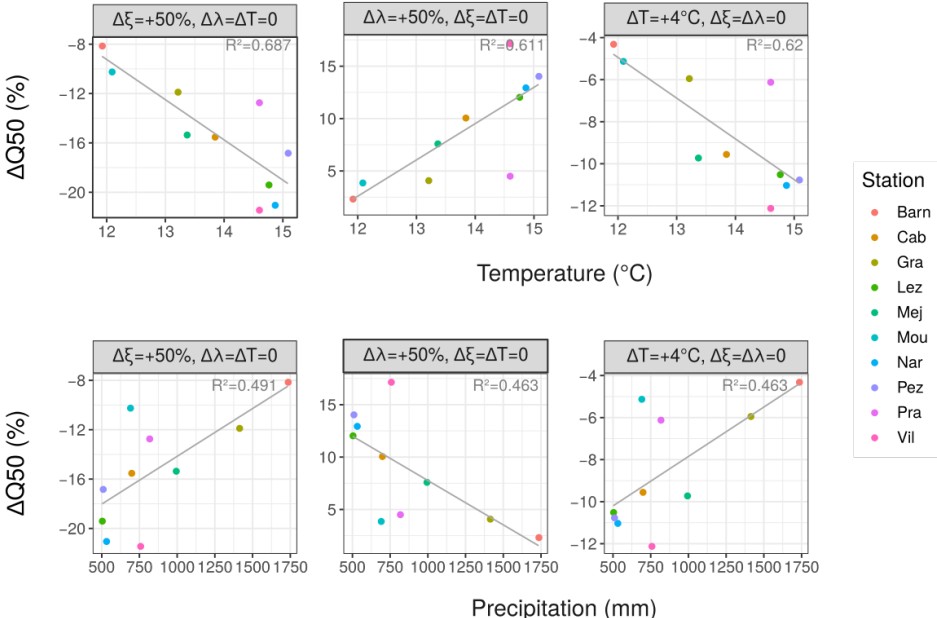

**Figure 7.** Sensitivity of the median of the simulated soil moisture to precipitation and temperature scenarios ($\Delta$: precipitation intensity scenario, $\Delta\lambda$: precipitation intermittence scenario, $\Delta T$: temperature scenario) related to the observed mean annual a) temperature b) precipitation.

The figure 8 presents for every station the 1st-order Sobol indices of the median soil moisture (resp. number of days under drought or excess condition) to the parameter change (temperature, precipitation intensity and precipitation intermittence). For instance, the Sobol index of the soil moisture to a parameter is the percentage of the soil moisture variance explained by the considered parameter. Over all the stations, the sum of the 1st-order Sobol indices are between 0.94 and 1.005 for
median soil moisture, between 0.92 and 0.99 for the number of days below the 5th percentile and between 0.93 and 0.99 for the days above the 95th percentile, which indicates that the GSA is based on a sufficient number of simulations. The Sobol sensitivity analysis shows that soil moisture variance is more impacted by changes in precipitation intermittence than changes in precipitation intensity and temperature, especially for the median soil moisture and the number of days with drought (i.e. low soil moisture values). Changes in precipitation intensities have a larger impact on higher soil moisture values and can became
almost equivalent to the changes in precipitation intermittency, as for example in the Pezenas station. There is a link with the mean precipitation and Sobol indices related to changes in precipitation intermittence and intensity. Indeed, the smaller the annual precipitation, the higher the Sobol index to the precipitation intermittence is for the median and 95th percentile of soil moisture (with correlations equal to respectively r=-0.71, r=-0.56). It is the opposite relationship between annual precipitation and precipitation intensity (with correlations equal to r=0.77 for median soil moisture, r=0.33 for the 5th percentile and r=0.74
for the 95th percentile). This indicates that changes in precipitation intermittence are more strongly impacting soil moisture in locations with low annual precipitation.



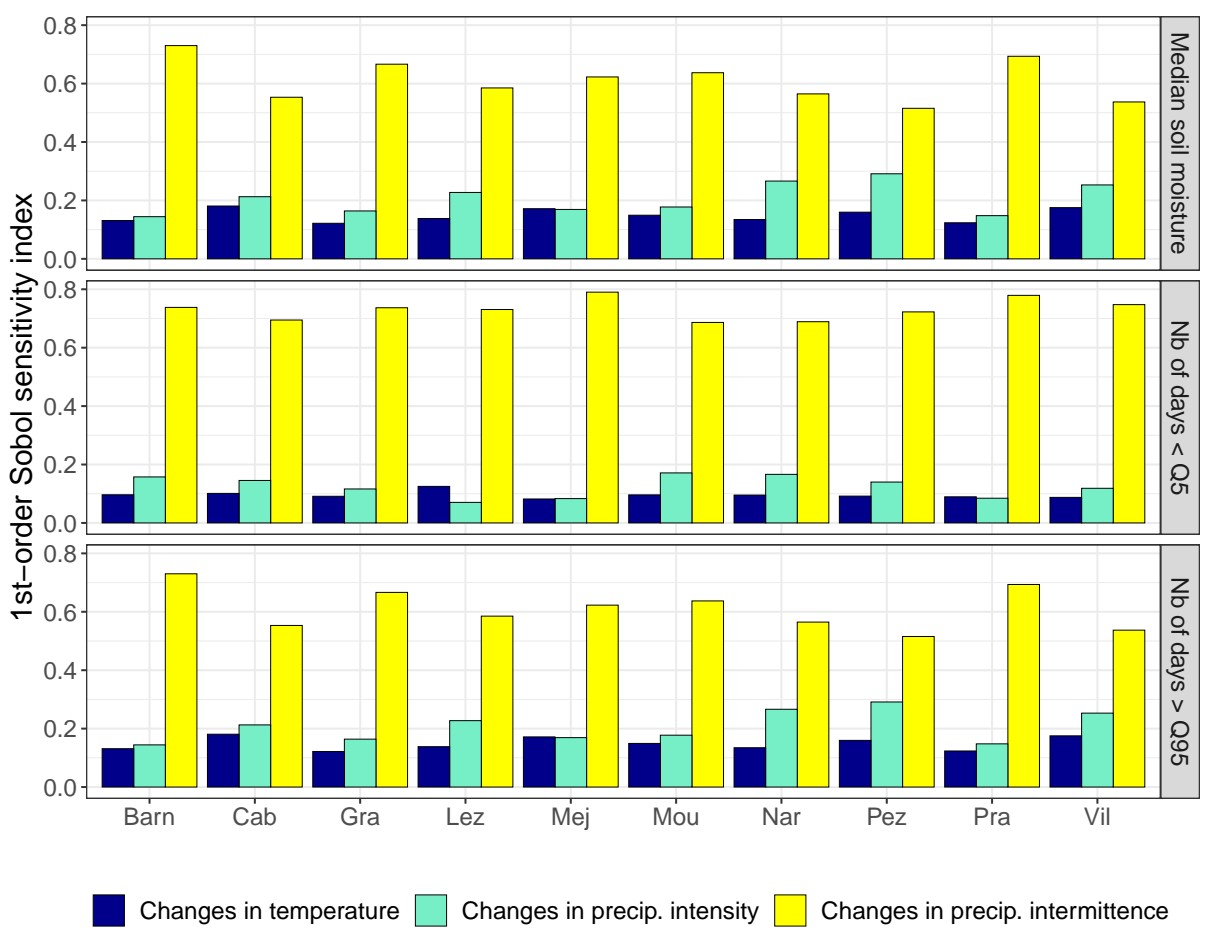

**Figure 8.** First order Sobol sensitivity index of median soil moisture (upper panel), the number of days under drought conditions (center panel), and the number of days with water excess (lower panel) to temperature, precipitation intensity and precipitation intermittence changes.

### 4.4 Impact of changing precipitation and temperature on extreme soil moisture

In this section we analyse the response of extreme soil moisture to the precipitation and temperature scenarios. Figure 9 shows the relative change of the mean annual number of days under saturation or drought conditions with respect to the reference scenario for the Barnas and Pezenas stations (complete results are presented in supplementary material). Days under saturation

5 (drought) conditions are defined as days with a daily soil moisture above the 95th (below the 5th) percentile of the reference scenario.

There is a large variability in the evolution of the mean annual number of days with wet conditions with results ranging from -14 to +12 days per year for the +2 °C scenario and from -15 to +8 days per year for the +4 °C scenario (Fig. 9a). For the +2 °C scenario, only 22 % of the scenarios result in an increase of annual days with wet conditions in average for the 11 stations. An

10 increase of annual wet days appears for scenarios showing a high increase in precipitation intensity (in average +37 %) and





moderate increase of precipitation intermittence (in average +7 %). On average, an increase of the precipitation intermittence above 26 % result in a decrease of the annual number of wet days, regardless the increase of precipitation intensity. Regarding the +4 °C scenario, only 12 % of the scenarios result in an increase of the annual number of wet days, and all scenarios with an increase of the precipitation intermittence above 17 % result in a decrease of the annual number of wet days. Scenarios similar

to the RCP8.5 scenario of Polade et al. (2017) (i.e. scenarios corresponding to a decrease of annual precipitation ranging between -10 and -20 % and a +4 °C temperature increase) lead to an average of 10 days per year with wet conditions, i.e. a decrease of 8 days per year relatively to the reference scenario (Fig 10).

Concerning the impact of changing precipitation and temperature on dry soil moisture conditions, Figure 9b shows that almost all scenarios lead to an increase of the mean annual number of dry days. For the +4 °C scenario, the increase of the

10 frequency of days with dry soil moisture can reach up to +40 days/yr for the Mazan and Narbonne stations. Only a few scenarios with a high increase of precipitation intensity and a low increase of precipitation intermittence result in decrease dry conditions (10 % for the +2 °C scenario, and 2.4 % for the +4 °C scenario). RCP8.5 scenarios show a mean number of dry days per year ranging between 34 and 52 days/yr, corresponding to a mean increase of +22 days per year comparing to the reference scenario (Fig 10). This increase of dry days mainly impacts the summer and autumn seasons from June to October (Fig. 11). None of

15 the stations show an increase of extreme dry days during winter. These results show that agricultural drought events in the Mediterranean region are likely to be more intense with longer episodes extending until the months of October and November.

Overall, results show that changes in precipitation patterns and temperature have a larger impact on lowest range of the soil moisture distribution than on the highest. This means that climate change is very likely to have a major impact on agricultural droughts with dryer soil moisture and longer drought events. Regarding the impact on flood events, it is difficult to make

conclusions based on the results of this study as we do not simulate runoff generation. Our results show a decrease of the median soil moisture for most of the considered scenarios as well as a decrease of days under saturated conditions suggesting a higher infiltration capacity of the surface soil layer with a potential lower runoff generation.

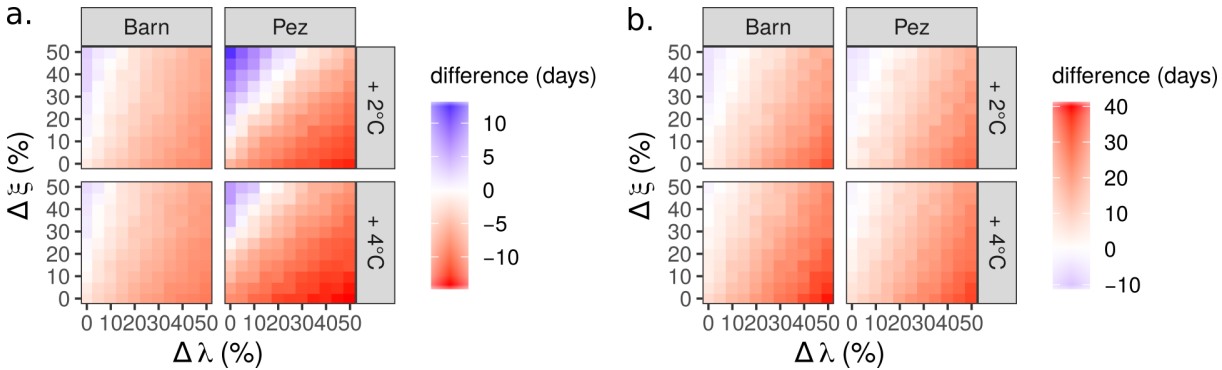

**Figure 9.** Sensitivity of the annual number of days (a) with saturated soil (i.e. with soil moisture above the observed 95th percentile) and (b) under extreme drought (i.e. with soil moisture below the observed 5th percentile), according to changes in precipitation intensity (y axis), precipitation intermittence (x axis) and temperature, for the Barnas and Pezenas stations.





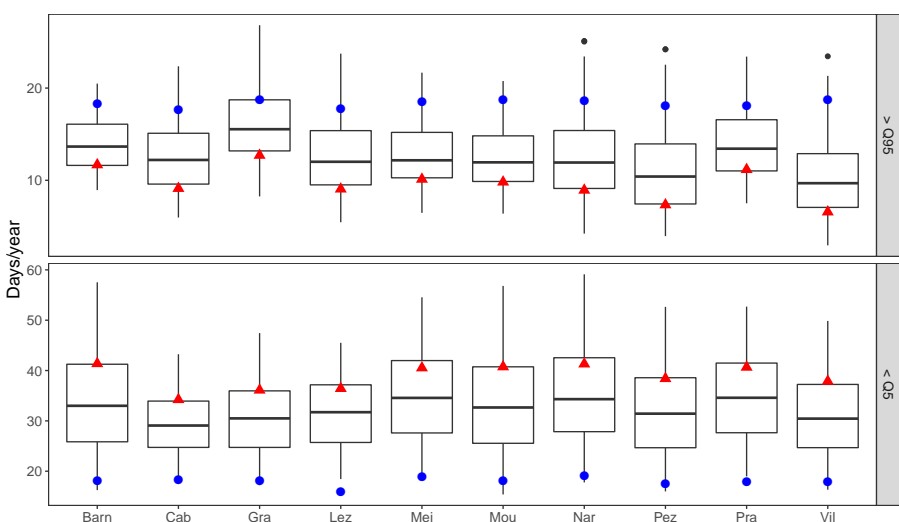

**Figure 10.** Variability of the annual number of days under saturated condition (SM above the observed 95th percentile, upper panel) and under extreme drought (SM below the observed 5th percentile, lower panel) at each station for a +4°C temperature scenario. Boxplots represent the results for all precipitation scenarios with increasing precipitation intensity and intermittence. Blue dots represent the reference scenario, with no change in temperature or precipitation pattern. Red triangles represent the mean of the scenarios with a decrease of annual precipitation between -10 and -20% (corresponding to scenario RCP8.5 Polade et al. (2017).





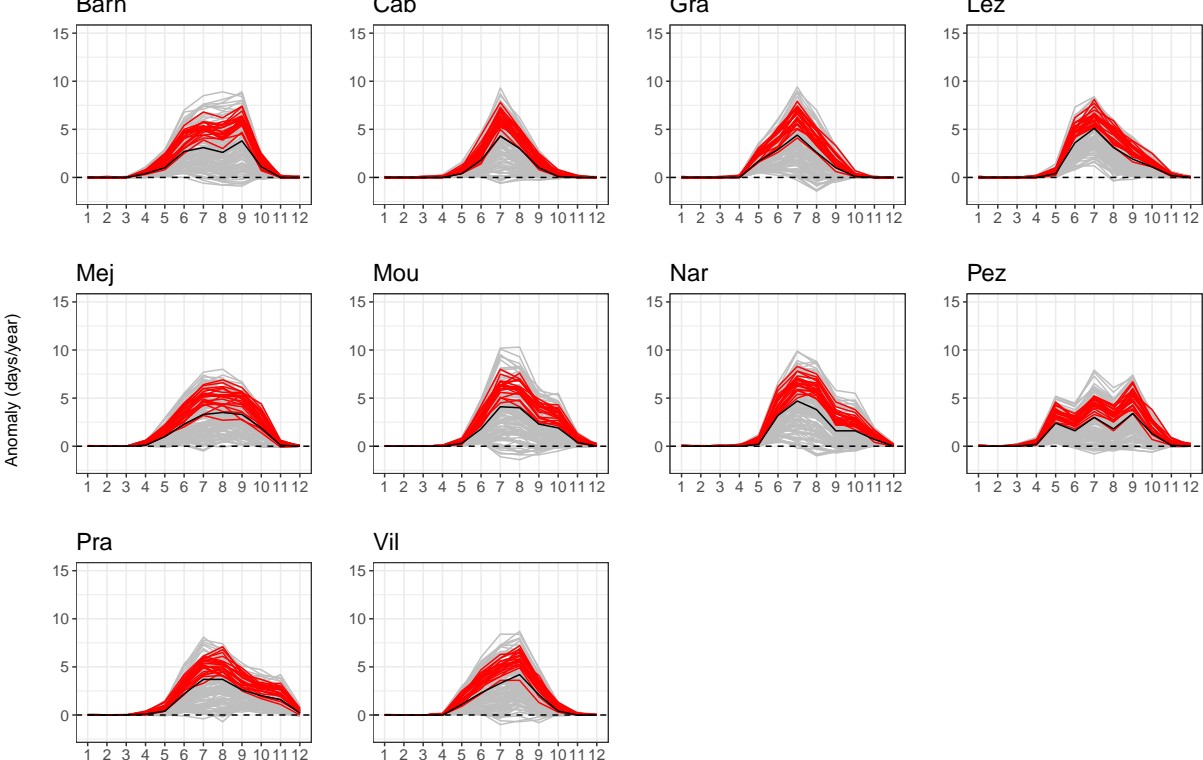

**Figure 11.** Monthly anomaly of days under extreme drought for a +4°C temperature scenario. Grey lines represent the results for all precipitation scenarios with increasing precipitation intensity and intermittence. Black lines represent the median of the scenarios ensemble. Red lines represent the change of drought days for the scenarios with a decrease of annual precipitation between -10 and -20% (corresponding to scenario RCP8.5 Polade et al. (2017).

## 5  Discussion

One of the main limitations to this study lies in the constant soil moisture model parameters under different climate scenarios. The use of constant parameters implies that processes such as the adaptation of vegetation to soil water stress or the impact of rising $CO_2$ on the vegetation physiology, which may have a sensitive impact on evapotranspiration and thus soil moisture (Berg and Sheffield, 2018), are not taken into account in this study. To avoid this issue, it would be required to consider land surface modelling schemes that are able to take into account the feedback effects between vegetation and land surface processes (Albergel et al., 2017). In addition, offline computation of potential evapotranspiration with standard formulas such as the Blaney and Criddle or Penman-Monteith equations can be problematic since it neglects several factors, in particular the surface conditions (Barella-Ortiz et al., 2013). The impact of different formulations of potential evapotranspiration on soil





moisture changes needs also to be investigated, since simple temperature-based formulas may overestimate the temperature effects on evapotranspiration (Sheffield et al., 2012; Vicente-Serrano et al., 2019).

Another source of uncertainties is related the selection of temperature and precipitation scenarios, while currently the majority of available climate simulations are at the daily time step. The projected changes on hourly climate characteristics remains largely unknown, and this is why we adopted a stochastic simulation approach to encompass the plausible range of future scenarios. However, convection-permitting regional climate models (CPRCM) are increasingly being implemented over Europe during the last years to reproduce hourly changes in precipitation (Coppola et al., 2018) and these simulations should be considered in future experiments. Similarly, the approach considered in the present paper is based on distributional changes, while the impact of possible changes in the seasonal to inter-annual variability of precipitations on soil moisture cannot be taken into account. This issue could be also resolved by using CPRCM simulations instead of a stochastic rainfall generator to simulate the soil moisture response to various changes in precipitation including seasonal and inter-annual variability.

Finally, this study relies on a set of soil moisture observations from different sites located in Southern France and, despite different annual precipitation and temperature patterns, the vegetation at the different locations belongs to the same biome. It would be interesting to perform this type of analysis on a larger set of sites located in various Mediterranean environments, including North Africa and the Middle East with more arid climate conditions, to investigate the possible relationships between soil moisture dynamics and soil types, vegetation cover and climate characteristics for different degrees of aridity. Indeed, the Mediterranean region includes a great variety of types of vegetation, forming mosaic patterns created by variations in soil, topography, climate, fire history and human activity (Geri et al., 2010). Therefore, it would be very useful to produce a typology of the sensitivity of soil moisture changes for a variety of Mediterranean landscapes.

## 6    Conclusions

Soil moisture is an important variable to consider in a climate change context since its strongly influences agricultural droughts and flood generation processes. Future climate scenarios for the Mediterranean indicate an increase in temperature, associated with an increased frequency of dry days but also an intensification of extreme rainfall events. This study considered soil moisture monitored at 10 plots located in southern France, in a modelling framework aiming at estimating its sensitivity to changes in precipitation and temperature. For that purpose, a range of precipitation and temperature variations coherent with current climate scenarios available for the Mediterranean region have been generated with a stochastic model to investigate the response of soil moisture to these climatic changes. The main result of this study shows that the sensitivity of soil moisture to changes in precipitation and temperature is similar at the different sites, with a higher sensitivity of soil moisture to intermittent precipitation and the number of dry days rather than their intensity or the temperature increase. However, these changes are modulated by the climate characteristics of the different stations, with a higher sensitivity of soil moisture to precipitation intermittence in locations with dryer and warmer climate characteristics. Overall, it is observed that changes in precipitation and temperature have a greater impact on low soil moisture values than on conditions close to soil saturation. This implies that the current climate change scenarios may induce longer periods of depleted soil moisture content, corresponding to agricultural





drought conditions. About the potential impacts of soil moisture changes on flood generation, more research is needed to better understand the joint influence of lower antecedent soil moisture conditions associated with higher rainfall intensity on flood magnitude and occurrence.

*Data availability.* The computed indices are available upon request to the corresponding author

5 **7 Supplementary materials**

| Code | From | To | Missing data [%] | | |
|------|------|-----|---------------|-------------|---------------|
| | | | Precipitation | Temperature | Soil moisture |
| Barn | 14/11/2008 | 31/12/2016 | 1.5% | 1.5% | 45% |
| Cab | 14/11/2008 | 31/12/2016 | 0.9% | 1.2% | 2.2% |
| Gra | 13/12/2008 | 31/12/2016 | 0.7% | 0.3% | 2.6% |
| Lez | 01/01/2007 | 31/12/2016 | 0.6% | 1.2% | 12.6% |
| Mej | 09/12/2008 | 31/12/2016 | 1.1% | 0.7% | 12.1% |
| Mou | 01/01/2007 | 31/12/2016 | 1.0% | 0.3% | 3.1% |
| Nar | 01/01/2007 | 31/12/2016 | 0.2% | 0.2% | 3.2% |
| Pez | 11/12/2008 | 18/04/2016 | 0.6% | 0.6% | 1.5% |
| Pra | 11/12/2008 | 31/12/2016 | 0.3% | 0.4% | 3.7% |
| Vil | 15/12/2008 | 31/12/2016 | 0.7% | 0.5% | 4.4% |

**Table 4.** Availability of observed data





NSRP calibrated parameters:

### Barnas (Barn)

| Month | λ [hr] | β [hr] | n | η [hr] | ξ [mm.hr$^{-1}$] |
|---|---|---|---|---|---|
| 1 | 105.1 | 10.1 | 4.07 | 3.2 | 1.0 |
| 2 | 105.5 | 9.3 | 4.68 | 3.8 | 1.5 |
| 3 | 107.7 | 13.5 | 3.97 | 3.8 | 1.0 |
| 4 | 108.6 | 10.3 | 4.30 | 4.1 | 1.2 |
| 5 | 105.1 | 10.2 | 4.08 | 4.2 | 1.0 |
| 6 | 102.6 | 8.8 | 4.63 | 2.0 | 1.0 |
| 7 | 89.4 | 3.2 | 3.65 | 0.1 | 33.9 |
| 8 | 54.9 | 1.2 | 5.36 | 0.0 | 34.5 |
| 9 | 665.7 | 8.3 | 28.22 | 0.4 | 14.4 |
| 10 | 91.4 | 10.7 | 7.52 | 4.3 | 1.1 |
| 11 | 87.9 | 11.2 | 9.33 | 3.1 | 1.6 |
| 12 | 312.1 | 50.0 | 5.55 | 10.0 | 1.1 |

### Cabrières d'Avignon (Cab)

| Month | λ [hr] | β [hr] | n | η [hr] | ξ [mm.hr$^{-1}$] |
|---|---|---|---|---|---|
| 1 | 369.3 | 27.0 | 18.19 | 0.9 | 1.6 |
| 2 | 189.9 | 1.4 | 1.98 | 0.0 | 82.8 |
| 3 | 158.3 | 4.0 | 15.52 | 0.1 | 4.8 |
| 4 | 104.8 | 8.4 | 4.62 | 1.4 | 1.2 |
| 5 | 113.3 | 9.8 | 2.36 | 3.0 | 1.0 |
| 6 | 83.8 | 2.6 | 0.86 | 3.3 | 2.4 |
| 7 | 85.8 | 0.2 | 1.69 | 0.0 | 9.8 |
| 8 | 115.0 | 8.2 | 1.87 | 3.2 | 0.9 |
| 9 | 107.9 | 8.6 | 3.45 | 3.5 | 1.0 |
| 10 | 134.1 | 36.8 | 2.49 | 0.6 | 11.9 |
| 11 | 99.7 | 18.9 | 6.33 | 2.1 | 1.2 |
| 12 | 117.0 | 19.3 | 2.68 | 2.5 | 1.0 |

### La Grand Combe (Gra)

| Month | λ [hr] | β [hr] | n | η [hr] | ξ [mm.hr$^{-1}$] |
|---|---|---|---|---|---|
| 1 | 109.1 | 9.0 | 3.24 | 4.0 | 1.0 |
| 2 | 103.3 | 11.4 | 4.94 | 4.4 | 1.1 |
| 3 | 104.8 | 6.0 | 4.20 | 3.3 | 1.1 |
| 4 | 98.1 | 7.7 | 6.19 | 2.6 | 1.1 |
| 5 | 254.9 | 17.2 | 20.77 | 0.1 | 15.4 |
| 6 | 129.8 | 9.6 | 4.03 | 0.4 | 6.3 |
| 7 | 98.6 | 0.3 | 1.84 | 0.2 | 8.3 |
| 8 | 213.9 | 7.7 | 6.48 | 0.6 | 4.3 |
| 9 | 103.5 | 9.0 | 5.03 | 3.9 | 1.1 |
| 10 | 90.8 | 9.4 | 7.69 | 3.3 | 1.2 |
| 11 | 90.7 | 11.0 | 7.88 | 3.5 | 1.2 |
| 12 | 385.2 | 48.5 | 5.90 | 10.0 | 1.1 |

### Lézignan (Lez)

| Month | λ [hr] | β [hr] | n | η [hr] | ξ [mm.hr$^{-1}$] |
|---|---|---|---|---|---|
| 1 | 90.2 | 3.5 | 0.87 | 2.7 | 2.2 |
| 2 | 107.3 | 16.2 | 3.79 | 1.6 | 1.0 |
| 3 | 118.5 | 21.8 | 2.01 | 3.1 | 1.5 |
| 4 | 228.7 | 13.1 | 15.40 | 0.5 | 2.6 |
| 5 | 121.6 | 6.8 | 1.90 | 2.5 | 1.8 |
| 6 | 71.3 | 1.2 | 0.03 | 0.1 | 26.3 |
| 7 | 308.8 | 5.5 | 7.16 | 0.5 | 1.7 |
| 8 | 302.1 | 2.8 | 6.54 | 0.1 | 21.2 |
| 9 | 160.8 | 1.1 | 1.44 | 0.1 | 20.5 |
| 10 | 126.0 | 50.0 | 0.76 | 9.2 | 2.2 |
| 11 | 110.2 | 12.9 | 3.22 | 3.4 | 1.0 |
| 12 | 255.8 | 12.9 | 11.51 | 0.8 | 1.0 |



### Méjannes-Le-Clap (Mej)

| Month | λ [hr] | β [hr] | n | η [hr] | ξ [mm.hr$^{-1}$] |
|---|---|---|---|---|---|
| 1 | 109.7 | 6.7 | 2.98 | 4.0 | 1.0 |
| 2 | 101.3 | 6.9 | 4.88 | 2.4 | 1.1 |
| 3 | 109.1 | 8.2 | 3.19 | 3.5 | 1.0 |
| 4 | 110.1 | 5.4 | 3.19 | 2.8 | 1.4 |
| 5 | 109.9 | 15.4 | 3.53 | 2.6 | 1.0 |
| 6 | 84.5 | 3.4 | 2.51 | 1.7 | 1.3 |
| 7 | 120.1 | 0.9 | 1.75 | 0.2 | 21.2 |
| 8 | 172.4 | 1.2 | 4.52 | 0.4 | 5.9 |
| 9 | 386.3 | 27.1 | 12.57 | 0.2 | 31.2 |
| 10 | 252.5 | 20.5 | 18.27 | 0.2 | 15.3 |
| 11 | 97.1 | 10.4 | 6.32 | 3.0 | 1.2 |
| 12 | 112.1 | 11.2 | 2.54 | 4.7 | 1.1 |

### Mouthoumet (Mou)

| Month | λ [hr] | β [hr] | n | η [hr] | ξ [mm.hr$^{-1}$] |
|---|---|---|---|---|---|
| 1 | 102.1 | 20.8 | 5.11 | 1.4 | 1.4 |
| 2 | 106.9 | 19.7 | 4.68 | 2.0 | 1.0 |
| 3 | 100.8 | 23.1 | 4.52 | 1.8 | 1.5 |
| 4 | 164.7 | 23.3 | 8.34 | 0.7 | 2.5 |
| 5 | 92.4 | 7.6 | 2.79 | 1.3 | 2.2 |
| 6 | 117.7 | 15.6 | 2.82 | 0.2 | 7.6 |
| 7 | 148.5 | 6.1 | 4.88 | 0.2 | 4.4 |
| 8 | 114.0 | 8.8 | 1.21 | 2.7 | 1.2 |
| 9 | 106.4 | 6.9 | 2.18 | 1.9 | 1.3 |
| 10 | 124.3 | 27.1 | 0.77 | 5.1 | 2.7 |
| 11 | 107.1 | 19.1 | 4.62 | 3.4 | 1.2 |
| 12 | 144.3 | 20.2 | 6.99 | 2.1 | 0.7 |

### Narbonne (Nar)

| Month | λ [hr] | β [hr] | n | η [hr] | ξ [mm.hr$^{-1}$] |
|---|---|---|---|---|---|
| 1 | 115.3 | 9.4 | 1.86 | 3.7 | 0.9 |
| 2 | 107.9 | 5.9 | 0.67 | 10.0 | 0.9 |
| 3 | 114.2 | 9.7 | 2.26 | 3.7 | 1.0 |
| 4 | 105.9 | 13.8 | 4.17 | 2.2 | 1.0 |
| 5 | 110.5 | 6.7 | 1.35 | 2.5 | 2.1 |
| 6 | 293.5 | 0.7 | 3.41 | 0.1 | 33.7 |
| 7 | 293.0 | 5.8 | 5.62 | 0.4 | 3.1 |
| 8 | 111.3 | 4.2 | 1.08 | 0.2 | 23.6 |
| 9 | 29.9 | 0.7 | 0.58 | 0.0 | 352.3 |
| 10 | 280.4 | 16.2 | 1.66 | 10.0 | 1.6 |
| 11 | 103.0 | 8.9 | 4.33 | 2.9 | 1.0 |
| 12 | 198.1 | 6.8 | 5.62 | 0.3 | 4.5 |

### Pezenas (Pez)

| Month | λ [hr] | β [hr] | n | η [hr] | ξ [mm.hr$^{-1}$] |
|---|---|---|---|---|---|
| 1 | 107.2 | 10.4 | 3.62 | 1.1 | 1.0 |
| 2 | 117.8 | 5.4 | 1.84 | 2.6 | 1.3 |
| 3 | 111.2 | 9.2 | 2.92 | 4.0 | 1.0 |
| 4 | 148.3 | 7.0 | 11.82 | 0.2 | 5.2 |
| 5 | 97.4 | 4.2 | 0.91 | 2.1 | 2.0 |
| 6 | 50.1 | 1.0 | 2.21 | 0.0 | 66.9 |
| 7 | 77.7 | 1.0 | 1.98 | 0.0 | 66.6 |
| 8 | 50.1 | 1.0 | 2.09 | 0.0 | 54.8 |
| 9 | 666.7 | 20.9 | 4.66 | 10.0 | 1.2 |
| 10 | 112.8 | 23.5 | 2.96 | 2.0 | 1.4 |
| 11 | 111.3 | 22.2 | 3.57 | 3.3 | 1.1 |
| 12 | 227.0 | 1.1 | 28.30 | 0.9 | 0.3 |





| | | Prades-Le-Lez (Pra) | | | | | | Villevieille (Vil) | | | |
|---|---|---|---|---|---|---|---|---|---|---|---|
| Month | $\lambda$ [hr] | $\beta$ [hr] | n | $\eta$ [hr] | $\xi$ [mm.hr$^{-1}$] | Month | $\lambda$ [hr] | $\beta$ [hr] | n | $\eta$ [hr] | $\xi$ [mm.hr$^{-1}$] |
| 1 | 82.3 | 2.2 | 7.09 | 0.0 | 16.6 | 1 | 83.1 | 3.2 | 1.96 | 2.1 | 1.3 |
| 2 | 116.5 | 21.5 | 2.20 | 2.7 | 2.2 | 2 | 79.0 | 2.5 | 0.93 | 2.4 | 2.7 |
| 3 | 200.4 | 2.0 | 2.72 | 10.0 | 0.9 | 3 | 110.3 | 18.0 | 3.73 | 2.7 | 1.2 |
| 4 | 102.9 | 10.9 | 2.62 | 1.0 | 3.9 | 4 | 104.0 | 10.1 | 4.47 | 2.2 | 1.0 |
| 5 | 111.7 | 7.5 | 2.75 | 2.3 | 1.1 | 5 | 246.0 | 10.8 | 11.42 | 0.1 | 12.9 |
| 6 | 160.8 | 4.5 | 4.31 | 0.2 | 11.3 | 6 | 108.5 | 3.0 | 2.37 | 0.2 | 15.0 |
| 7 | 130.5 | 1.6 | 1.72 | 0.5 | 5.6 | 7 | 114.4 | 9.9 | 1.98 | 2.2 | 1.1 |
| 8 | 307.5 | 1.7 | 2.22 | 1.0 | 9.4 | 8 | 412.2 | 1.9 | 4.77 | 0.1 | 100.0 |
| 9 | 126.2 | 5.5 | 1.87 | 4.1 | 1.8 | 9 | 106.8 | 7.0 | 1.63 | 0.7 | 13.1 |
| 10 | 523.0 | 11.5 | 15.76 | 0.2 | 26.5 | 10 | 105.0 | 11.1 | 4.31 | 3.4 | 1.0 |
| 11 | 289.8 | 28.7 | 12.29 | 0.3 | 10.9 | 11 | 101.3 | 12.1 | 5.76 | 1.9 | 1.3 |
| 12 | 112.2 | 19.7 | 2.54 | 2.8 | 1.4 | 12 | 118.9 | 11.0 | 1.20 | 3.8 | 1.9 |

| Nom_station | soil class USDA | USDA range of $K_s$ | Calibrated $K_s$ |
|---|---|---|---|
| Bar | Sandy loam | 51 - 152 | 38 |
| Cab | Loam | 15 - 51 | 34 |
| Gra | Sandy loam | 51 - 152 | 36 |
| Lez | Loam | 15 - 51 | 23 |
| Mej | Loam | 15 - 51 | 29 |
| Mou | Clay loam | 5 - 15 | 36 |
| Nar | Clay | 2 - 5 | 51 |
| Pez | Loam | 15 - 51 | 15 |
| Pra | Clay loam | 5 - 15 | 60 |
| Vil | Sandy loam | 51 - 152 | 7 |

**Table 5.** Range of hydraulic conductivity values (min and max) based on soil class (Angerer et al., 2014)





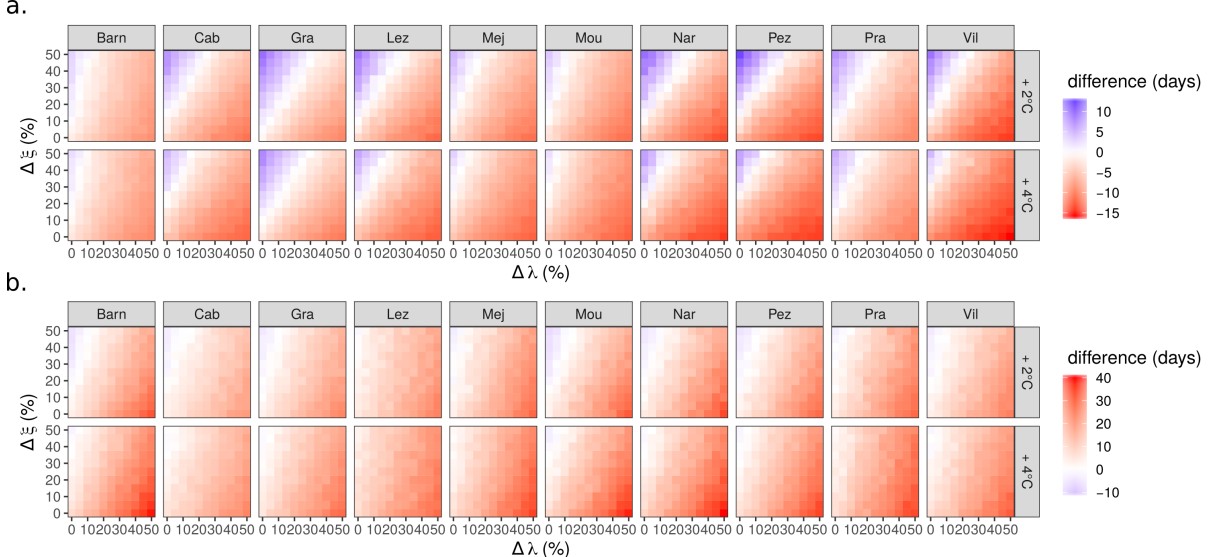

**Figure 12.** Sensitivity of the annual number of days with saturated soil (i.e. with soil moisture above the observed 95th percentile) according to changes in precipitation intensity (y axis), precipitation intermittence (x axis) and temperature. b. Sensitivity of the annual number of days under extreme drought (i.e. with soil moisture below the observed 5th percentile)

LM and YT designed the experiments, performed the analyses and wrote the paper, LB, CM and SC contributed to soil moisture modelling and climate scenarios, P F-G contributed to the sensitivity analysis. All authors helped the interpretation of results and revised the paper.

5 *Competing interests.*

Nothing to declare.

*Acknowledgements.* This work is a contribution to the HYdrological cycle in The Mediterranean EXperiment (HyMeX) program, through INSU-MISTRALS support. The authors would like to thank Météo-France for providing precipitation and temperature data, the soil moisture from the SMOSMANIA network has been downloaded from the International Soil Moisture Network (https://ismn.geo.tuwien.ac.at/en/).



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
