# Peer review of "Modeling the response of soil moisture to climate variability in the Mediterranean region"

_Hydrology and Earth System Sciences, 2020_

## Referee Comment (RC1) · Guillaume Evin (Referee) · 20 Jul 2020

I thank the authors for this interesting paper on the relationship between meteorological forcings and soil moisture in the Mediterranean region. The manuscript is well written, well organized and the different modelling tools are adequately applied. The first important result is that the increase in temperature is not the main driver of the changes in soil moisture, but seems to be precipitation characteristics. The second important contribution is methodological since this study shows how a soil moisture model and meteorological scenarios can be used to assess the sensitivity of the soil moisture to these forcings. I have two major comments (see below) regarding how rainfall scenarios are generated. The authors simulate changes of intermittency using the parameter lambda of the Neyman-Scott model. This lambda parameter is the master Poisson process parameter and is directly related to the frequency of rainfall events. I think that the interpretation of 'intermittence' is misleading, which is annoying since the main results of the paper rely on this interpretation. My main recommendation is thus to change the way rainfall scenarios are generated. In my opinion, the best option for the generation of scenarios would be to recalibrate the NSRP model for each set of rainfall statistics (the observed ones + the perturbed ones). In the current version of the manuscript, it must be clearly understood that when one parameter (e.g. lambda) is modified, it affects all rainfalls statistics, which complicates the interpretation of the main factors leading to changes in the soil moisture.

Major comments:

**1 Due to its structure, the different parameters of the Neyman-Scott rectangular pulse model are not directly interpretable in terms of rainfall statistics. In the current version of the manuscript, parameters lambda and xi are loosely interpreted in terms of "intermittence" and "mean intensity". In my opinion, this interpretation is incorrect and misleading: - The parameter lambda, which governs the master Poisson process, represents the rate of rainfall events (storms). As such, the mean intensity (for any aggregation duration) is linear in lambda (Eq. 2.5 in Cowpertwait, 1998). It is also true for the covariance for any lag (Eq. 2.6 in Cowpertwait, 1998). This means that when lambda decreases (in this paper the inverse of the storm frequency), the mean rainfall intensity increases in proportion. - The parameter xi is the parameter of the exponential distribution for raincell intensity. The mean rainfall intensity (for any aggregation duration) is linear in lambda. When xi increases, the mean rainfall intensity increases in proportion (Eq. 2.5 in Cowpertwait, 1998). An augmentation of 50% in lambda is directly compensated by an augmentation of 50% in xi, which is indicated in Section 4.1 (l. 20). However, an increase of xi with the same increase in lambda leads to the same annual rainfall but also to an increase of the mean intensity of the rainy days (which is indicated at l. 21 but not clearly since the authors refer to the "mean rainfall intensity"), and to an increase of the number of dry days. - Intermittency is not clearly**

defined in the paper. I strongly suggest proposing a definition in terms of rainfall statistics. A stronger intermittence could be, for the same annual rainfall, a higher number of dry days. It could be parametrized with lambda and xi, but also with the other parameters. Note also that the theoretical proportion of dry days can be easily obtained with the NSRP model (see Eq. 9a-9b in Cowpertwait, 1991), using a numerical integration. The two quantities that would be perturbed could thus be "the total annual rainfall" and "the proportion of dry days" (or equivalently the number of dry days), which would have a direct interpretation. - As said above, in my opinion, the only valid option for the generation of scenarios is to recalibrate the NSRP model for each set of rainfall statistics (the observed ones + the perturbed ones). When lambda or xi is modified, it affects many rainfalls statistics at the same time, which complicates the interpretation of the main factors leading to changes in the soil moisture. As the proportion of dry days is important in this study, it should also be included in the set of rainfall statistics used to estimate the parameters. Cowpertwait, Paul S. P. 1991. "Further Developments of the Neyman-Scott Clustered Point Process for Modeling Rainfall." Water Resources Research 27 (7): 1431–38. https://doi.org/10.1029/91WR00479. Cowpertwait, Paul S. P. 1998. "A Poisson-Cluster Model of Rainfall: High-Order Moments and Extreme Values." Proceedings: Mathematical, Physical and Engineering Sciences 454 (1971): 885–98.

**2 Many parameter estimates seem to indicate a failure of the estimation method. For eta, the raincell duration parameter, many zero values appear (e.g. Pezenas, June to August) associated to very high values of xi and 1 for beta (the initial value of the optimization I guess). In Pezenas, in September, eta reaches the highest value of 10 I guess, and lambda is very high (666.7). It affects maybe 10 months for all the stations, but the problem should be addressed. I cannot trust these simulations with these unrealistic parameter estimates. Possible solutions are:**

1. Try different starting values for the optimization, 2. Change the objective functions (weighted sums, relative/absolute differences between observed and simulated statistics), 3. Smooth the estimation from one month to another, there is no strong reason to have a big difference between two consecutive months.

Minor comments: p.2, l.14: Repetition of "soil moisture", "For soil moisture" could be removed. p.2, l.32: For your information, a recent reference of scenario neutral approaches is "Keller, Luise, Ole Rössler, Olivia Martius, and Rolf Weingartner. 2019. "Comparison of Scenario-Neutral Approaches for Estimation of Climate Change Impacts on Flood Characteristics." Hydrological Processes 33 (4): 535–50. https://doi.org/10.1002/hyp.13341". p.3, l.5: with -> and. Figure 1: Missing labels (Longitude / Latitude). Figure 2: Please increase the font size. p.7, l.3: "of" should be removed. p.7, l.4-8: I think it would be fair to indicate that it is the standard version of the NSRP model, many more elaborate versions have been proposed in the last decades. p.7: The mean number of raincell per storm is often denoted by the Greek letter nu, as is actually done in the manuscript in Section 3.4.1. p.8, l.1: I suggest indicating the statistical properties used for the estimation of the parameters. For these statistics at least, we should have a good agreement between the observations and the simulated values. p.8, l.11: +4C the symbol "degree" is missing. p.10, l.4: there is a space after "+4" that should be removed. p.10, l.11: There is a slight overestimation of the annual number of dry days for some stations (e.g. Barn), it could be noticed. p.12: m3.m-3 seems to be a strange unit (adimensional actually), is it correct?

---

## Referee Comment (RC2) · Anonymous Referee #2 · 27 Jul 2020

I am very excited to see a manuscript that (finally) shows changing rainfall intermittency is important for soil moisture and hence flooding. This is very important and timely work. I can only expect (and look forward to) seeing the follow up work to this study involving the impact on flooding. Please see my minor (and bordering on pedantic) suggestions below.

I am not sure on the format of HESS – but the "Annexe" references didn't quite match the SI for me. I am not sure I saw a reference to the full calibration parameters?

Page 1, Line 6: "on a 10 year time period" -> "for a 10 year time period"

Page 2, Line 4: I would appreciate Wasko and Nathan (2019) to be cited alongside these as, though similar to Bennett et al (2018), it goes beyond to quantify the impact

of soil moisture changes with flood recurrence.

Wasko, C., & Nathan, R. (2019). Influence of changes in rainfall and soil moisture on trends in flooding. Journal of Hydrology, 575, 432–441. https://doi.org/10.1016/j.jhydrol.2019.05.054

Page 1, Line 9: Just wonder if the following also supports the drying trend you are referring to.

Rodell, M., Famiglietti, J. S., Wiese, D. N., Reager, J. T., Beaudoing, H. K., Landerer, F. W., & Lo, M. H. (2018). Emerging trends in global freshwater availability. Nature, 557(7707), 651–659. https://doi.org/10.1038/s41586-018-0123-1

Page 1, Line 20: Obviously I am more familiar with Australian references, but the following evaluates in-situ soil moisture. I would make the point that one issue with evaluation is the different depths that are measured and modelled by all products.

Holgate, C. ., De Jeu, R. A. M., van Dijk, A. I. J. ., Liu, Y. ., Renzullo, L. J., Vinodkumar, et al. (2016). Comparison of remotely sensed and modelled soil moisture data sets across Australia. Remote Sensing of Environment, 186, 479–500. https://doi.org/10.1016/j.rse.2016.09.015

Page 3, Line 2: Please have a look at the following as I think it is also looking at soil moisture using a scenario-neutral approach:

Stephens, C. M., Johnson, F. M., & Marshall, L. A. (2018). Implications of future climate change for event-based hydrologic models. Advances in Water Resources, 119, 95–110. https://doi.org/10.1016/j.advwatres.2018.07.004

Figure 1: If it isn't too much of a hassle it would be nice to see Figure 1 include an inset of the study site in the context of the greater region (as I am not familiar with the study region). But this is only a suggestion and I don't mind if this isn't performed.

Page 7, Line 2: "series"

Page 8, Line 5: I recognize studies often change all the parameters in the NSRP model for downscaling (e.g. Bordoy and Burlando, 2014). If you are looking for examples of where a parameter is fixed in stochastic generation based on, for example, physical intuition you can see Wasko et al (2015) and Onof and Wheater (1994).

Bordoy, R., & Burlando, P. (2014). Stochastic downscaling of climate model precipitation outputs in orographically complex regions: 2. Downscaling methodology. Water Resources Research, 50(1), 562–579. https://doi.org/10.1002/wrcr.20443

Wasko, C., Pui, A., Sharma, A., Mehrotra, R., & Jeremiah, E. (2015). Representing low-frequency variability in continuous rainfall simulations: A hierarchical random Bartlett Lewis continuous rainfall generation model. Water Resources Research, 51(12), 9995–10007. https://doi.org/10.1002/2015WR017469

Onof, C., & Wheater, H. S. (1994). Improvements to the modelling of British rainfall using a modified random parameter Bartlett-Lewis rectangular pulse model. Journal of Hydrology, 157, 177–195. https://doi.org/http://dx.doi.org/10.1016/0022-1694(94)90104-X

Page 8, Line 14: "resumes" -> "presents"

Page 8, Line 21: where you say the modelling chain is processed 20 times, I think you mean to say "stochastic replicates" or "simulated ensembles" – this terminology I think is clearer.

Page 9, Line 8: A typo has occurred. Remove "the )"?

Section 4.1: These increases of say 432mm, is this for one site in particular? Or across all the sites on average? I am a bit confused here.

Page 10, Line 9, 12: "The" NSRP model?

Page 10, Line 18: "Opposite" -> "Alternatively"

Page 15, Line 1: "The Figure" -> "Figure"

[Figure]

Page 15, Lin 9: "became" -> "be"

Figure 9, 11, 12 captions: I think these say "extreme" drought while in other parts of the manuscript you just say "drought". I would stick to the terminology "drought".

Page 17, Line 7: Can you mention in the text what the blue and red symbols in Figure 10 are and maybe specifically mention how the RCP changes predicted are at the "extreme" ends of your scenario space. If I have interpreted the results correctly his point was lost on me but is very important to highlight I think?

Page 21, Line 2: The following manuscript is one of the few manuscripts demonstrating how drier soils interact with higher precipitation intensities.

Wasko, C., & Nathan, R. (2019). Influence of changes in rainfall and soil moisture on trends in flooding. Journal of Hydrology, 575, 432–441. https://doi.org/10.1016/j.jhydrol.2019.05.054

---

## Referee Comment (RC3) · Ryan Teuling (Referee) · 31 Jul 2020

The manuscript by Mimeau et al. addresses the important issue of changes in soil moisture conditions in the Mediterranean. The stochastic approach is a nice addition to existing studies, and the main findings are important. The topic also fits very well in the special issue. However I have some concerns regarding details in the Methods, the use of literature on stochastic approaches to soil moisture dynamics, and the presentation of the results. These are discussed below. I believe the concerns are best addressed in a major revision.

Introduction

"Only a few studies attempted to validate the soil moisture simulated by the GCM or

RCM land surface schemes" -> Maybe, but other studies (such as Stegehuis, GRL, 2013, doi.org/10.1002/grl.50404) have used flux observations which should have the same, if not better, effect.

"This is particularly true for the Mediterranean regions ... land surface models" -> Ok, but next you claim this can be solved by using a simplified model. So are the other models all worse than the simple model used here? Or is the lack of calibration of higher importance than model structure?

"The only study that applied this method to soil moisture" -> There are at least several others, such as Teuling et al. (GRL 2007, doi:10.1029/2007GL031001), and Calanca et al. (WRR 2004, doi:10.1029/2004WR003254)

Literature: In general, I miss a discussion on the previous use of stochastic approaches in soil moisture modeling. These include for instance the work by Milly (WRR 2011, doi:10.1029/2000WR900337), Laio et al. (AWR 2001, 24, 707-723), and Rodriguez-Iturbe (1999, Proc R Soc Lond A 455: 3789-805). These (analytical) approaches use a more basic description of the precipitation process, so it should be motivated why a more complex Neyman-Scott representation is needed to address the research question.

Method

Table 2 mentions the "Monthly potential evaporation coefficient L". What is the role of this parameter, and how is it different from the coefficient for evapotranspiration Kc?

"a linear relationship between actual and potential evapotranspiration" -> Please provide more information. Is this linear between field capacity and wilting point? If so, this is a big simplification. Many other studies have shown that there is a considerable range in soil moisture over which ET is potential (above the critical moisture content), and that this unstressed soil moisture range is in fact required to explain observed soil moisture and vegetation dynamics and features such as strong bimodality (Salvucci,

2001 WRR 37(5), 1357–1365, Teuling et al. GRL 2005 doi:10.1029/2005GL023223, Denissen et al. JGR 2020, doi:10.1029/2019JD031672). It should be better motivated why this gross simplification is justified, and what the potential implications are for the simulated soil moisture dynamics (for instance, the higher stress could explain why most lines in Fig5 are above the 1:1 line around 20 Vol%)

"two additional calibrations were performed on subperiods . . . in order to analyze the stability of the calibration" -> For the stability it is more important to consider the variability in optimum parameters than the model performance itself (that is listed in Table 3). Please also provide the parameters for periods 1 and 2 so that the robustness of the calibration can be better assessed.

In the method, it is mentioned that the rainfall parameters are estimated for each month of the year. I assume that this also means that the model is run for every month separately? This is not mentioned. If so, this has some implications for the results, because in this way one doesn't account for the month-to-month carry-over of soil moisture memory (i.e. going into summer the soil moisture will be slightly higher at the beginning of each month because of the on average wetter previous month). Please explain and discuss the potential impacts this approach has on the results.

Results

I miss an illustration of model performance, for instance a modeled and simulated time-series at one of the stations so that model performance can be visually checked (NSE tends to be high by default in strongly seasonal climates, so this alone might not be a good indication).

Figure 8: This is an important figure, but I find it difficult to extract any relevant information other than that intermittence is the most sensitive factor. This could more easily be shown by first averaging over all stations, and only show the stations if there a story to it. The most important aspect now is the comparison between the different rows, and this is not easy because the reader has to guess the values and compare visually.

Consider plotting the differences more explicit if this is where conclusions are based on.

---

## Author Comment (AC1) · 16 Oct 2020

**Reviewer 1, Guillaume Evin**

**I thank the authors for this interesting paper on the relationship between meteorological forcings and soil moisture in the Mediterranean region. The manuscript is well written, well organized and the different modelling tools are adequately applied. The first important result is that the increase in temperature is not the main driver of the changes in soil moisture, but seems to be precipitation characteristics. The second important contribution is methodological since this study shows how a soil moisture model and meteorological scenarios can be used to assess the sensitivity of the soil moisture to these forcings. I have two major comments (see below) regarding how rainfall scenarios are generated. The authors simulate changes of intermittency using the parameter lambda of the Neyman-Scott model. This lambda parameter is the master Poisson process parameter and is directly related to the frequency of rainfall events. I think that the interpretation of 'intermittence' is misleading, which is annoying since the main results of the paper rely on this interpretation. My main recommendation is thus to change the way rainfall scenarios are generated. In my opinion, the best option for the generation of scenarios would be to recalibrate the NSRP model for each set of rainfall statistics (the observed ones + the perturbed ones). In the current version of the manuscript, it must be clearly understood that when one parameter (e.g. lambda) is modified, it affects all rainfalls statistics, which complicates the interpretation of the main factors leading to changes in the soil moisture.**

Thank you for this in-depth review of the manuscript and in particular of the stochastic generation method applied in our work.

We acknowledge the concerns raised by the approach considered. While it is true that some authors applied this re-calibration of the rainfall generator after the modification of rainfall statistics (Burlando and Rosso, 2002; Bordoy and Burlando, 2014), other studies considered a similar approach as ours, by modifying directly the parameters of interest in the rainfall generator (Onof and Wheater, 1994; Wasko et al., 2015).

We tested the approach proposed, based on the recalibration of the generator, but this approach did not provide satisfactory results. We also improved the calibration of the rainfall generator. Please find a more detailed response below.

Burlando, P. and Rosso, R.: Effects of transient climate change onbasin hydrology, 1. Precipitation scenarios for the Arno River Basin, central Italy, Hydrol. Process, 16, 1151–1175, 2002.

Bordoy, R. and Burlando, P.: Stochastic downscaling of climate model precipitation outputs in orographically complex regions: 2. Downscaling methodology. Water Resources Research, 50(1), 562–579. https://doi.org/10.1002/wrcr.20443, 2014

**Major comments:**

**#1 Due to its structure, the different parameters of the Neyman-Scott rectangular pulse model are not directly interpretable in terms of rainfall statistics. In the current version of the manuscript, parameters lambda and xi are loosely interpreted in terms of "intermittence" and "mean intensity". In my opinion, this interpretation is incorrect and misleading: - The parameter lambda, which governs the master Poisson process, represents the rate of rainfall events (storms). As such, the mean intensity (for any aggregation duration) is linear in lambda (Eq. 2.5 in Cowpertwait, 1998). It is also true for the covariance for any lag (Eq. 2.6 in Cowpertwait, 1998). This means that when lambda decreases (in this paper the inverse of the storm frequency), the mean rainfall intensity increases in proportion. - The parameter xi is the parameter of the exponential distribution for rain cell intensity. The mean rainfall intensity (for any aggregation duration) is linear in lambda. When xi increases, the mean rainfall intensity increases in proportion (Eq. 2.5 in Cowpertwait, 1998). An augmentation of 50% in lambda is directly compensated by an augmentation of 50% in xi, which is indicated in Section4.1 (l. 20). However, an increase of xi with the same increase in lambda leads to the same annual rainfall but also to an increase of the mean intensity of the rainy days (which is indicated at l. 21 but not clearly since the authors refer to the "mean rainfall intensity"), and to an increase of the number of dry days. - Intermittency is not clearly defined in the paper. I strongly suggest proposing a definition in terms of rainfall statistics. A stronger intermittence could be, for the same annual rainfall, a higher number of dry days. It could be parametrized with lambda and xi, but also with the other parameters. Note also that the theoretical proportion of dry days can be easily obtained with the NSRP model (see Eq. 9a-9b in Cowpertwait, 1991), using a numerical integration. The two quantities that would be perturbed could thus be "the total annual rainfall" and "the proportion of dry days" (or equivalently the number of dry days), which would have a direct interpretation. - As said above, in my opinion, the only valid option for the generation of scenarios is to recalibrate the NSRP model for each set of rainfall statistics (the observed ones + the perturbed ones). When lambda or xi is modified, it affects many rainfalls statistics at the same time, which complicates the interpretation of the main factors leading to changes in the soil moisture. As the proportion of dry days is**

**important in this study, it should also be included in the set of rainfall statistics used to estimate the parameters. Cowpertwait, Paul S. P. 1991. "Further Developments of the Neyman-Scott Clustered Point Process for Modeling Rainfall." Water Resources Research 27 (7): 1431–38. https://doi.org/10.1029/91WR00479. Cowpertwait, Paul S. P. 1998. "A Poisson-Cluster Model of Rainfall: High-Order Moments and Extreme Values." Proceedings: Mathematical, Physical and Engineering Sciences 454 (1971):885–98.**

We implemented the approach proposed, by first modifying the rainfall statistics, and then recalibrating the rainfall generator based on the modified rainfall statistics.

Figure 1 shows that the characteristics of the generated rainfall time series after the perturbation of the rainfall statistics and recalibration are not consistent with the perturbation of the observed rainfall statistics. For instance, for an increase in precipitation intensity and no change in precipitation intermittence, some stations (Lez, Nar, Pez, Vil) do not show any increase in total precipitation (Fig 1 upper panel).

And regarding the impact on soil moisture, Figure 2 shows that, with this method, an increase in precipitation intensity leads to a decrease in the median soil moisture for most of the stations, which seems unrealistic.

[Figure]

Fig 1: Change in annual precipitation (upper panel), daily rainfall intensity (middle panel), and annual number of dry days (lower panel) obtained after perturbating the rainfall statistics and recalibrating the rainfall generator.

[Figure]

Fig 2: Sensitivity of the median of the simulated soil moisture to an increase of the mean daily rainfall intensity (left panel), and to an increase of mean number if dry days (right panel) under different temperature scenarios (+0 °C, +2 °C, +4 °C)

Consequently, since this approach is not satisfactory in our case, most probably due to the interdependence of different parameters in the Neyman-Scott model (as noted by the reviewer) that is probably amplified when conducting many re-calibration procedures, we kept the initial approach of perturbing the rainfall generator parameters.

But we added Figures 3 and 4 into the manuscript to show the relation between the perturbation of the parameters λ and ξ the change in the number of dry days and precipitation intensity of the generated rainfall series. Figure 3 shows that the perturbation of the parameter ξ is equivalent to perturbating the mean rainfall intensity. There is also a clear relation between the modified value of the λ parameter and the mean number of dry days. An increase of 100% of the λ parameter leads to an increase ranging between 10 and 18 % of the number of dry days depending on the station.

[Figure]

Figure 3: Change in the number of dry days (left panel) and rainfall intensity (right panel) when perturbing the λ and ξ parameters of the rainfall generator.

[Figure]

Figure 4: Change in the rainfall characteristics (upper panel: mean annual precipitation, middle panel: mean daily rainfall intensity, lower panel: mean number of dru days) of the generated rainfall time series when increasing the λ and ξ parameters from 0 to +100%.

In the initial manuscript we considered perturbations of the parameters up to +50%, which were equivalent to an increase up to 50% of the mean daily intensity and an increase up to 10% of the mean number of dry days. These values might be in fact too low to analyse the impact of extreme changes in rainfall patterns to soil moisture, that is why we extended the range of perturbation of the parameters up to +100%.

All the figures of the manuscript were updated in the revised manuscript. The general results and main conclusions remain the same as the ones in the submitted manuscript.

**#2 Many parameter estimates seem to indicate a failure of the estimation method. For eta, the rain cell duration parameter, many zero**

**values appear (e.g. Pezenas, June to August) associated to very high values of xi and 1 for beta (the initial value of the optimization I guess). In Pezenas, in September, eta reaches the highest value of10 I guess, and lambda is very high (666.7). It affects maybe 10 months for all the stations, but the problem should be addressed. I cannot trust these simulations with these unrealistic parameter estimates. Possible solutions are: 1. Try different starting values for the optimization, 2. Change the objective functions (weighted sums, relative/absolute differences between observed and simulated statistics), 3. Smooth the estimation from one month to another, there is no strong reason to have a big difference between two consecutive months**

We tried applying different starting values for the optimization, with a monthly variation to have a smoother variability between two consecutive months. Results show that the different calibration strategies we tested in order to modify the initial values for the calibration do not significantly reduce the unrealistic parameters values obtained and give very similar results in terms of rainfall intensities.

We kept in the revised manuscript the calibration results that yielded the most realistic values for the rainfall generator parameters. We also checked carefully the rainfall intensities generated to make sure we did not produce strongly biased values. The figure 5 below shows very similar rainfall intensities obtained with the different calibration strategies.

[Figure]

Figure 5: Density plot of observed (green) and simulated (red) hourly rainfall intensities

**Minor comments:**

**p.2, l.14: Repetition of "soil moisture", "For soil moisture" could be removed.**

removed

**p.2, l.32: For your information, a recent reference of scenario neutral approaches is "Keller, Luise, Ole Rössler, Olivia Martius, and Rolf Weingart-ner. 2019. "Comparison of Scenario-Neutral Approaches for Estimation of ClimateChange Impacts on Flood Characteristics." Hydrological Processes 33 (4): 535–50.https://doi.org/10.1002/hyp.13341".**

we added this reference page 2, line 32

**p.3, l.5: with -> and.**

Replaced

**Figure 1: Missing labels (Lon-gitude / Latitude).**

Added

[Figure]

Figure 6: Localisation of the study sites in southern France

**Figure 2: Please increase the font size.**

Modified

**p.7, l.3: "of" should be removed.**

Removed

**p.7, l.4-8: I think it would be fair to indicate that it is the standard version of the NSRP model, many more elaborate versions have been proposed in the last decades.**

We added "the standard version of .." "

**p.7: The mean number of raincell per storm is often denoted by the Greek letter nu, as is actually done in the manuscript in Section 3.4.1.**

We replaced by Greek nu to be consistent with section 3.4.1.

**p.8, l.1: I suggest indicating the statistical properties used for the estimation of the parameters. For these statistics at least, we should have a good agreement between the observations and the simulated values.**

We added the rainfall properties:
"The statistical properties of rainfall included in the objective function to calibrate the model are: hourly mean, hourly variance, daily variance, lag1 autocorrelation of daily data, hourly skewness, daily skewness and the percentage of dry days."

**p.8, l.11: +4C the symbol "degree" is missing.**

Added

**p.10, l.4: there is a space after "+4" that should be removed.**

Removed

**p.10, l.11: There is a slight overestimation of the annual number of dry days for some stations (e.g. Barn), it could be noticed.**

Added

**p.12:m3.m-3 seems to be a strange unit (adimensional actually), is it correct**

It is the standard unit for soil moisture measurements, so it is also the unit of the RMSE values computed.

---

## Author Comment (AC2) · 16 Oct 2020

**Reviewer 2**

**I am very excited to see a manuscript that (finally) shows changing rainfall intermittency is important for soil moisture and hence flooding. This is very important and timely work. I can only expect (and look forward to) seeing the follow up work to this study involving the impact on flooding. Please see my minor (and bordering on pedantic) suggestions below.**

We would like to thank you for your positive appraisal of our manuscript. Please find below the answer to your comments.

**I am not sure on the format of HESS – but the "Annexe" references didn't quite match the SI for me.**

We modified the format of supplementary materials.

**I am not sure I saw a reference to the full calibration parameters?**

We added page 10, line 14: "(see supplementary material S2 for the calibrated NSRP parameters)".

**Page 1, Line 6: "on a 10 year time period" -> "for a 10 year time period"**

Changed

**Page 2, Line 4: I would appreciate Wasko and Nathan (2019) to be cited along side these as, though similar to Bennett et al (2018), it goes beyond to quantify the impact of soil moisture changes with flood recurrence.**

We added this reference page 2 line 4

**Page 1, Line 9: Just wonder if the following also supports the drying trend you are referring to. Rodell, M., Famiglietti, J. S., Wiese, D. N., Reager, J. T., Beaudoing, H. K., Landerer,F. W., & Lo, M. H. (2018). Emerging trends in global freshwater availability. Nature,557(7707), 651–659. https://doi.org/10.1038/s41586-018-0123-1**

Indeed, but this paper refers to groundwater dynamics, where in this section we are mentioning trends in atmospheric water supply.

**Page 1, Line 20: Obviously I am more familiar with Australian references, but the following evaluates in-situ soil moisture. I would**

**make the point that one issue with evaluation is the different depths that are measured and modelled by all products.**
**Holgate, C. ., De Jeu, R. A. M., van Dijk, A. I. J. ., Liu, Y. ., Renzullo, L. J., Vin-odkumar, et al. (2016). Comparison of remotely sensed and modelled soil moisture data sets across Australia. Remote Sensing of Environment, 186, 479–500.https://doi.org/10.1016/j.rse.2016.09.015**

We added this reference page 2, line 22. The issue of evaluation at different depths is mostly valid for remote sensing data, able to measure soil moisture for the surface only, when in climate models the land surface scheme is capable of reproducing also the root zone soil moisture.

**Page 3, Line 2: Please have a look at the following as I think it is also looking at soil moisture using a scenario-neutral approach: Stephens, C. M., Johnson, F. M., & Marshall, L. A. (2018). Implications of future climate change for event-based hydrologic models. Advances in Water Resources, 119, 95–110.**

We added this reference page 2, line 31.

**Figure 1: If it isn't too much of a hassle it would be nice to see Figure 1 include an inset of the study site in the context of the greater region (as I am not familiar with the study region). But this is only a suggestion and I don't mind if this isn't performed.**

We added a small map of France in the top left corner to locate the region of interest (see our response to reviewer 1, Figure 6)

**Page 7, Line 2: "series"**

Changed

**Page 8, Line 5: I recognize studies often change all the parameters in the NSRP model for downscaling (e.g. Bordoy and Burlando, 2014). If you are looking for examples of where a parameter is fixed in stochastic generation based on, for example, physical intuition you can see Wasko et al (2015) and Onof and Wheater (1994).**
**Bordoy, R., & Burlando, P. (2014). Stochastic downscaling of climate model precipita-tion outputs in orographically complex regions: 2. Downscaling methodology. WaterResources Research, 50(1), 562–579. https://doi.org/10.1002/wrcr.20443**
**Wasko, C., Pui, A., Sharma, A., Mehrotra, R., & Jeremiah, E. (2015). Representing low-frequency variability in continuous rainfall simulations: A hierarchical ran-dom Bartlett Lewis continuous rainfall**

generation model. Water Resources Research,51(12), 9995–10007. https://doi.org/10.1002/2015WR017469

Onof, C., & Wheater, H. S. (1994).Improvements to the modelling of Britishrainfall using a modified random parameter Bartlett-Lewis rectangular pulse model.Journal of Hydrology, 157, 177–195. https://doi.org/http://dx.doi.org/10.1016/0022-1694(94)90104-X

Following the recommendations of Reviewer 1, Guillaume Evin, we also tested a different approach by recalibrating all the parameters of the rainfall generator after modifying the rainfall statistics (similar to Bordoy and Burlando 2014), see our response. Since this approach did not work well, we kept our original approach, similar to the two references you mentioned. We added the two references proposed.

**Page 8, Line 14: "resumes" -> "presents"**

Changed

**Page 8, Line 21: where you say the modelling chain is processed 20 times, I think you mean to say "stochastic replicates" or "simulated ensembles" – this terminology I think is clearer.**

Changed to "a simulated ensemble of 20 stochastic replicates is generated"

**Page 9, Line 8: A typo has occurred. Remove "the )"?**

Changed

**Section 4.1: These increases of say 432mm, is this for one site in particular? Or across all the sites on average? I am a bit confused here.**

We added: across all sites on average

**Page 10, Line 9, 12: "The" NSRP model?**

Added

**Page 10, Line 18: "Opposite" -> "Alternatively"**

Changed

**Page 15, Line 1: "The Figure" -> "Figure"**

Changed

**Page 15, Lin 9: "became" -> "be"**

Changed

**Figure 9, 11, 12 captions: I think these say "extreme" drought while in other parts of the manuscript you just say "drought". I would stick to the terminology "drought".**

We removed "extreme" in the figure captions

**Page 17, Line 7: Can you mention in the text what the blue and red symbols in Figure10 are and maybe specifically mention how the RCP changes predicted are at the "extreme" ends of your scenario space. If I have interpreted the results correctly his point was lost on me but is very important to highlight I think?**

We added this information about the symbols in the text page 17, line 7 and 8.

**Page 21, Line 2: The following manuscript is one of the few manuscripts demonstrating how drier soils interact with higher precipitation intensities. Wasko, C., & Nathan, R. (2019).Influence of changes in rainfall andsoil moisture on trends in flooding.Journal of Hydrology, 575, 432–441.https://doi.org/10.1016/j.jhydrol.2019.05.054**

We added this reference in the introduction.

---

## Author Comment (AC3) · 16 Oct 2020

**Reviewer 3: Ryan Teuling**

**The manuscript by Mimeau et al. addresses the important issue of changes in soil moisture conditions in the Mediterranean. The stochastic approach is a nice addition to existing studies, and the main findings are important. The topic also fits very well in the special issue. However I have some concerns regarding details in the Methods, the use of literature on stochastic approaches to soil moisture dynamics, and the presentation of the results. These are discussed below. I believe the concerns are best addressed in a major revision.**

Thank you for the revision of our manuscript. Please find below a point-by-point response to your comments and the modifications made to the revised manuscript.

**Introduction "Only a few studies attempted to validate the soil moisture simulated by the GCM or RCM land surface schemes" -> Maybe, but other studies (such as Stegehuis, GRL,2013, doi.org/10.1002/grl.50404) have used flux observations which should have the same, if not better, effect.**

We note that in Stegehuis et al 2013, there is no evaluation of simulated soil moisture but only sensible heat flux at the surface and the 2-m mean temperature.

**"This is particularly true for the Mediterranean regions . . . land surface models" -> Ok, but next you claim this can be solved by using a simplified model. So are the other models all worse than the simple model used here? Or is the lack of calibration of higher importance than model structure?**

We do not claim that the large variability between climate model simulations of soil moisture can be solved with a simpler model than the land-surface schemes of the climate models. We just state that there are obvious discrepancies in soil moisture simulated by these different models, so we prefer to rely on a bottom-up approach based on observed data to estimate the sensitivity of soil moisture to changes in climate characteristics. We added: "As a consequence, the direct use of soil moisture from climate models may not be the best option to assess small scale soil moisture variability in relation with climate conditions."

**"The only study that applied this method to soil moisture" -> There are at least several others, such as Teuling et al. (GRL 2007, doi:10.1029/2007GL031001), and Calanca et al. (WRR 2004, doi:10.1029/2004WR003254)**

**Literature: In general, I miss a discussion on the previous use of stochastic approaches in soil moisture modeling. These include for instance the work by Milly (WRR 2011,doi:10.1029/2000WR900337), Laio et al. (AWR 2001, 24, 707-723), and Rodriguez-Iturbe (1999, Proc R Soc Lond A 455: 3789-805). These (analytical) approaches use a more basic description of the precipitation process, so it should be motivated why a more complex Neyman-Scott representation is needed to address the research question.**

Thank you for these additional references. We modified this section to include the proposed references, and provide a better review of previous studies applying stochastic methods to soil moisture. We do not think that a complex stochastic generator is necessarily required. For instance, Zhu et al 2020 used a rather simple elasticity approach or Guo et al 2018 applied a simpler weather generator to achieve satisfactory results. We used a Newman-Scott model to represent distinctly the changes in precipitation intermittence and intensity at the hourly time step but other approaches can be equally valid too, as soon as they are able to represent changes in these rainfall properties.

**Method**
**Table 2 mentions the "Monthly potential evaporation coefficient L". What is the role of this parameter, and how is it different from the coefficient for evapotranspiration Kc?**

L is the monthly percentage of total daytime hours out of total daytime hours of the year. This fixed parameter, computed based on the station's coordinates, enables to represent the monthly variations of the potential evaporation. The values of L are not calibrated.

Kc is a correction factor that is calibrated for each station to adjust evapotranspiration.

Brocca, L., Camici, S., Melone, F., Moramarco, T., Martínez-Fernández, J., Didon-Lescot, J.-F., & Morbidelli, R. (2013). *Improving the representation of soil moisture by using a semi-analytical infiltration model. Hydrological Processes, 28(4), 2103–2115.* doi:10.1002/hyp.9766

**"a linear relationship between actual and potential evapotranspiration" -> Please provide more information. Is this linear between field capacity and wilting point? If so, this is a big simplification. Many other studies have shown that there is a considerable range in soil moisture over which ET is potential (above the critical moisture content), and that this unstressed soil moisture range is in fact required to explain observed soil moisture and vegetation dynamics and features such as strong**

**bimodality (Salvucci, 2001 WRR 37(5), 1357–1365, Teuling et al. GRL 2005 doi:10.1029/2005GL023223,Denissen et al. JGR 2020, doi:10.1029/2019JD031672). It should be better motivated why this gross simplification is justified, and what the potential implications are for the simulated soil moisture dynamics (for instance, the higher stress could explain why most lines in Fig5 are above the 1:1 line around 20 Vol%)**

This was an error in the model description. It is not a linear relation between actual and potential evapotranspiration but a linear relation between potential evapotranspiration and soil saturation, that is used to compute actual evapotranspiration. See equation 7 of Brocca et al., 2008. This formulation is quite standard and many models use it.

We modified the text to remove this error, thank you for noticing this mistake.

**"two additional calibrations were performed on subperiods . . . in order to analyze the stability of the calibration" -> For the stability it is more important to consider the variability in optimum parameters than the model performance itself (that is listed in Table3). Please also provide the parameters for periods 1 and 2 so that the robustness of the calibration can be better assessed.**

We added the calibrated parameters (soil moisture model) on the 2 periods, see table below. It can be seen that the parameters values are within the same order of magnitude for the three calibration periods, with a stronger variability of the Ks compared to the two other parameters.

| | Barn | Cab | Gra | Lez | Mej | Mou | Nar | Pez | Pra | Vil |
|---|---|---|---|---|---|---|---|---|---|---|
| | Calibration on the total period | | | | | | | | | |
| $K_s$ (mm.hr$^{-1}$) | 38.1 | 34.3 | 35.9 | 23.1 | 28.8 | 36.2 | 51.1 | 14.6 | 59.6 | 6.9 |
| $m$ | 17.6 | 15.6 | 10.9 | 14.1 | 16.4 | 23.0 | 15.9 | 12.8 | 11.89 | 38.2 |
| $K_c$ | 1.17 | 1.43 | 1.74 | 1.22 | 1.81 | 0.94 | 1.26 | 1.99 | 1.32 | 1.63 |
| NSE | 0.76 | 0.77 | 0.93 | 0.85 | 0.9 | 0.63 | 0.91 | 0.789 | 0.65 | 0.9 |
| | Calibration on the first sub-period | | | | | | | | | |
| $K_s$ (mm.hr$^{-1}$) | 26.9 | 52.0 | 56.2 | 41.6 | 24.6 | 22.5 | 52.0 | 24.0 | 61.9 | 22.6 |
| $m$ | 17.8 | 15.8 | 11.3 | 15.4 | 14.7 | 17.8 | 17.5 | 21.0 | 10.5 | 40.0 |
| $K_c$ | 1.28 | 1.49 | 1.95 | 1.30 | 1.76 | 1.02 | 1.31 | 1.86 | 1.31 | 1.63 |
| NSE | 0.6 | 0.72 | 0.86 | 0.87 | 0.80 | 0.69 | 0.87 | 0.31 | 0.64 | 0.87 |
| | Calibration on the second sub-period | | | | | | | | | |
| $K_s$ (mm.hr$^{-1}$) | 29.6 | 23.7 | 43.9 | 40.8 | 45.6 | 77.4 | 43.1 | 6.4 | 71.4 | 2.7 |
| $m$ | 23.2 | 17.1 | 11.3 | 14.8 | 18.9 | 22.4 | 13.5 | 5.5 | 13.0 | 39.9 |
| $K_c$ | 1.09 | 1.42 | 1.53 | 1.11 | 1.87 | 1.32 | 1.19 | 1.97 | 1.31 | 1.56 |
| NSE | 0.71 | 0.75 | 0.87 | 0.78 | 0.91 | 0.04 | 0.912 | 0.60 | 0.57 | 0.86 |

**Table 3.** Calibrated parameters of the SM model and NSE validation values while calibrating on the total period, the first and second sub-periods of the in situ data series.

**In the method, it is mentioned that the rainfall parameters are estimated for each month of the year. I assume that this also means that the model is run for every month separately? This is not mentioned. If so, this has some implications for the results, because in this way one doesn't account for the month-to-month carry-over of soil moisture memory (i.e. going into summer the soil moisture will be slightly higher at the beginning of each month because of the on average wetter previous month). Please explain and discuss the potential impacts this approach has on the results.**

The NSRP model (the rainfall generator) is applied to each month separately, since the distribution of rainfall needs to be homogeneous (the distribution of hourly rainfall is obviously not the same in December or in August in these Mediterranean areas). This is a very standard practice when using this type of rainfall generator. If the distributions are estimated for each month, the generator then simulates continuous rainfall series, to be used as inputs in the soil moisture model and provide time series across all months/years.

We added page 8, line 2: "Once the model parameters estimated for each month, it is run to produce continuous simulations.".

**Results**

**I miss an illustration of model performance, for instance a modeled and simulated time-series at one of the stations so that model performance can be visually checked (NSE tends to be high by default in strongly seasonal climates, so this alone might not be a good indication).**

We added a new figure with the observed and simulated time series of soil moisture.

[Figure]

Simulated (green) and observed (red) soil moisture at the Villevieille station

**Figure 8: This is an important figure, but I find it difficult to extract any relevant information other than that intermittence is the most sensitive factor. This could more easily be shown by first averaging over all stations, and only show the stations if there a story to it. The most important aspect now is the comparison between the different rows, and this is not easy because the reader has to guess the values and compare visually. Consider plotting the differences more explicit if this is where conclusions are based on.**

The fact that intermittence is the key factor is indeed the main message of this figure. We modified the figure according to your recommendation, showing boxplots of the Sobol indices for Temperature (Temp.), Precipitation intensity (Pr. Intens.) and Precipitation Intermittence (Pr. Inter.), see the new figure below:

[Figure]

[Figure]

[Figure]